# Ellipsoidal Time Series Forecasting

**Qilin Wang** [1]

## Abstract

We argue that long-horizon forecasting requires learning local Jacobians with explicit spectral structure, not only matching conditional means. Our method, Fern (**F**orecasting with **E**llipsoidal **R**epresentatio**N**s), invokes Brenier's theorem to directly parameterize the Jacobian as a symmetric positive semi-definite (SPD) factorization, treating forecasting as the optimal transport of probability mass from a fixed Gaussian source to data-dependent ellipsoids. This formulation avoids post-hoc eigendecomposition of dense Jacobians, enables efficient Householder-based orthogonal factors, and exposes interpretable diagnostics such as local stretching, spectral radius, and volume change. To rigorously evaluate robustness, we introduce controlled synthetic stress tests with nonstationary shocks, together with Wasserstein-based shape metrics and Effective Prediction Time. Fern demonstrates exceptional stability, outperforming baselines like DLinear and Koopa by over two orders of magnitude (up to $790\times$) on nonstationary settings where standard benchmarks fail to expose model brittleness.

## 1. Introduction

We view long-horizon time series forecasting (LTSF) as *conditional manifold transport*: given a context window such as $[x_1, \ldots, x_{70}]$, an evenly sampled *time-delay embedding* (TDE), can one locate the current position on the underlying manifold and transport probability mass along its local geometry to future horizons, e.g., $[y_1 := x_{71}, \ldots, x_{100}]$? Truly long-horizon prediction is possible on simple, stable geometry such as a sine wave: given current position, traversing the wave through time gives the conditional prediction. Consider a non-trivial example where geometric perspective is essential: chaos. Recovering the canonical

Lorenz-63 (Lorenz, 1963)'s tightly coupled equations purely from data is already hard; even *knowing* the equations is *insufficient*. Due to chaos's *sensitive dependence on initial conditions*, the time for nearby trajectories to diverge by a factor of e (the Lyapunov time) is remarkably short. Any microscopic observation error amplifies *exponentially*, inevitably causing pointwise trajectory tracking to fail. Yet, dissipative chaotic systems evolve toward *attractors*, sets of states that retain coherent geometric structure. Between the post-Lyapunov and pre-mixing regimes, pointwise prediction fails, but geometric accuracy remains meaningful: a forecast can lose exact phase alignment with the true trajectory while still remaining close to the correct invariant geometry. Therefore, LTSF must go beyond simple time-stepping and explicitly model manifold transport.

Real systems add further complications. *Noise* blurs the manifold; at high noise levels, little invariant structure survives. *Nonstationary shocks* can induce regime switches, partitioning the data into piecewise-ergodic segments with distinct local signatures. In this view, forecasting requires tracking not a single global geometry, but a collection of local geometries that may change across regimes. In domains such as finance, stochastic noise, deterministic chaos, and nonstationary regime shifts can coexist, compounding the difficulty. These observations motivate a local, data-dependent view of forecasting. Rather than seeking a single global evolution rule, we model the conditional map around each context window, allowing different regions and regimes to induce different local geometries. Importantly, local geometries must provide diagnostics to shed light on the failure modes of the model.

*Direct Jacobian modeling* is consistent with the data-driven philosophy of *The Bitter Lesson* (Sutton, 2019): rather than imposing hand-crafted priors such as trend-seasonal decompositions, we *generate* local linearized maps from context window. Such maps expose rich geometric diagnostics without committing to a full manifold reconstruction: eigenvalues and eigenvectors of the Jacobian describe directional stretching and contraction. The difficulty is computational. A dense Jacobian over an $n$-dimensional forecast horizon has $n^2$ entries, and analyzing its spectrum by eigendecomposition adds $O(n^3)$ cost. For large horizons, this makes direct Jacobian modeling difficult without either reducing capacity or reverting to autoregressive generation.

[1]Independent Researcher. Correspondence to: Qilin Wang <841129@gmail.com>.

*Proceedings of the 43rd International Conference on Machine Learning*, Seoul, South Korea. PMLR 306, 2026. Copyright 2026 by the author(s).

**Attractor Reconstruction (Side-by-Side)**

Ground Truth (Color: Speed)                                    Prediction (Color: Eigen Max)

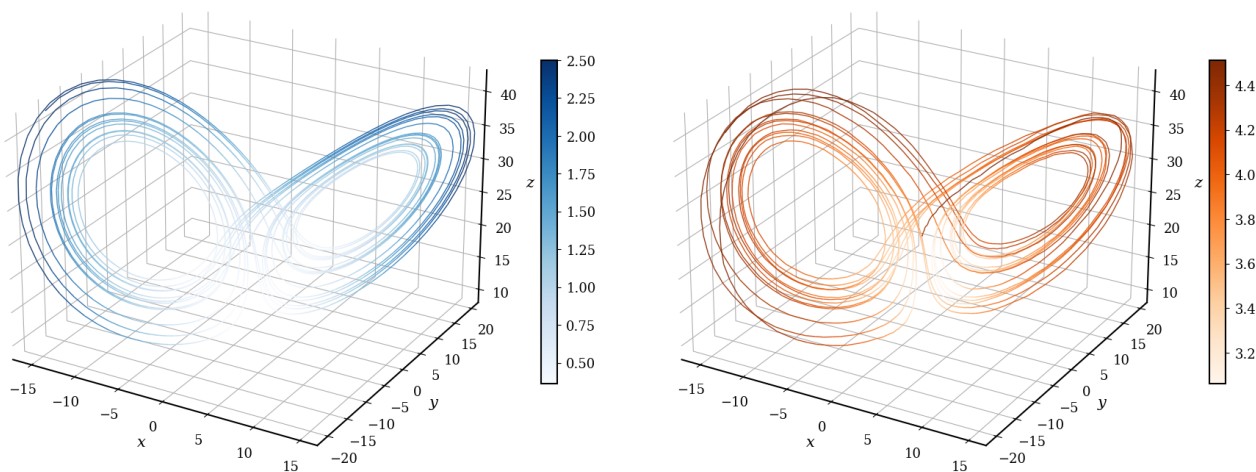

*Figure 1.* Lorenz-63 attractor reconstruction with Fern. Left: ground-truth trajectory colored by instantaneous speed (norm of velocity). Right: Fern prediction, colored by the mean patch-wise maximum eigenvalue (spectral radius of the local SPD map).

Our method, Fern, parameterizes forecasting as a conditional transport from a fixed Gaussian source in forecast space to a target distribution selected by the context $x$. This is not an OT map from $x$-space to $y$-space: the context selects the map, while the transported object is Gaussian source noise. This formulation addresses three bottlenecks. First, by restricting the predictive map to the *symmetric positive semi-definite (SPD)* class, Fern searches over spectral factors rather than arbitrary dense Jacobians, without losing the ability to target arbitrary Gaussian conditional distributions within each patch. With $R$ Householder reflections, the orthogonal factor can be applied in $O(Rn)$ rather than the $O(n^2)$ cost of a dense Jacobian. Second, Fern directly parameterizes the spectrum, avoiding the $O(n^3)$ eigendecomposition that would be required if a dense Jacobian were learned first and analyzed afterward. Third, the construction applies patchwise, so multiple lower-dimensional transports can be predicted in parallel with shared backbones. For example, a 336-step horizon can be represented by fourteen 24-dimensional patch maps rather than one monolithic 336-dimensional Jacobian — a $14\times$ reduction, since $14 \cdot 24^2 \ll 336^2$.

Because all patches transport from a common Gaussian source $y_0 \sim \mathcal{N}(0, I)$, their eigenvalues are directly comparable: a large eigenvalue in one patch and another in a different patch both measure how much the model stretches the same isotropic reference to reach its local prediction. The spectral output thus reads as a diagnostic signal: large eigenvalues mark regions where the model believes large local stretching is required, small ones mark stable regions. In the Lorenz-63 system (Fig. 1), ground truth (left) acceler-

ates along the outer rings (deep blue) and decelerates at the "bottleneck" (white). Remarkably, under simple Huber-loss training with no supervision on eigenvalues, Fern's maximum eigenvalues (right) spike in the high-speed regions (dark orange). The model discovers, purely from minimizing point error, that large stretching is required where the system moves fastest. Standard probabilistic forecasting reads uncertainty off explicit scoring objectives (NLL, CRPS); here the diagnostic falls out of the spectral parameterization as a byproduct of point-loss training.

## 2. Methods

The structural motivation for Fern is Brenier's theorem (Brenier, 1991). For each context window $x$, consider a conditional target distribution $\eta_x$ over the forecast space $\mathcal{Y}$. Rather than transporting $x$ itself, Fern transports a reference variable $y_0 \sim \nu = \mathcal{N}(0, I)$ within $\mathcal{Y}$ to a forecast distribution conditioned on $x$. Under regularity conditions that $\nu$ is absolutely continuous and $\eta_x$ has finite second moments, Brenier's theorem states that there exists a unique Wasserstein-2 optimal map $G$ from $\nu$ to $\eta_x$ i.e. $G_\sharp \nu = \eta_x$ that minimizes the moving cost $\int \|u - G(u)\|_2^2 \, d\nu(u)$; this map is the gradient of a convex potential. Hence its Jacobian is *symmetric positive semi-definite (SPD)* almost everywhere.

In the Gaussian case (think of wrapping the conditional mean of $y \mid x$ with Gaussian errors as in regression), this structure becomes explicit. The target conditional distribution is approximated as $\mathcal{N}(\mu(x), \Sigma(x))$; the *learned* condi-

tional Gaussian

$$q_\theta(y \mid x) = \mathcal{N}(\mu_\theta(x), \Sigma_\theta(x)),$$

and the optimal affine transports from $\mathcal{N}(0, I)$ to these two Gaussians are, respectively,

$$y = G(y_0; x) = \mu(x) + A(x)y_0$$
$$\hat{y} = G_\theta(y_0; x) = \mu_\theta(x) + A_\theta(x)y_0$$

The Brenier maps select the unique SPD square roots ($A(x)$ and $A_\theta(x)$) so that $A(x)A(x)^\top = \Sigma(x)$ and $A_\theta(x)A_\theta(x)^\top = \Sigma_\theta(x)$.

Three structural facts follow. First, Brenier's theorem guarantees a unique W2-optimal map from $\mathcal{N}(0, I)$ to *any* Gaussian target distribution, and the SPD map class—the *existence class*—is sufficiently expressive. Second, Fern's affine SPD parameterization means that every emitted pair $(\mu_\theta(x), \Sigma_\theta(x))$ defines the exact W2-optimal transport from Gaussian source noise to its own learned Gaussian distribution. Thus the map is OT-exact by construction, not only at convergence; training determines which learned Gaussian is selected, not whether the emitted Gaussian has an exact OT map from the source. Third, since each OT is *conditional on* a context window $x$, the OT between source and prediction is *instance-wise* (or *fibre-wise*). It does not transport the distribution of $x$ itself. This is distinct from global, *marginal* OT between empirical laws over context windows and future windows.

This setup grants key flexibility: under point losses such as MSE, Huber, or MAE, the supervised target is the conditional location parameter, not the true conditional covariance: MSE identifies the conditional mean, MAE the conditional median, and Huber a robust intermediate. For symmetric distributions, including the Gaussian, these coincide. For example, $\mathcal{N}(\mu^\star(x), \Sigma)$ and $\mathcal{N}(\mu^\star(x), 3\Sigma)$ share the same optimal point prediction under any of these losses. In such cases, the learned covariance is *an internal spectral belief state* shaped by point-loss training; the parameterization remains valid as an exact OT map even though $\Sigma_\theta$ is not supervised. If the goal is for $\Sigma_\theta(x)$ to approximate the true conditional second-order structure, Fern can instead be trained with an explicit probabilistic or distributional objective, such as likelihood, CRPS, or an OT-based loss.

Brenier's theorem applied to patches allows further cost cuts: let $p$ be the patch size and $n_p = n/p$ the number of patches. There is an OT map between the 70-dim $\nu$ and the 70-dim $\eta_x$, but there equally exists a *different* OT between the first 10-dim patch of source and the first 10-dim patch of target, since least-costly transports differ between overall and local problems. A full-capacity patch map costs $O(p^2)$, giving total cost $O(n_p p^2) = O(np)$ across patches, while a reduced Householder map costs $O(Rp)$ per patch and $O(n_p Rp) = O(Rn)$ overall. Since each patch's prediction

does not depend on the predictions of other patches, the patchwise transport runs in parallel. This is not a relaxation, but a modeling *parameterization* choice: one large 70-dim output-side transport, or many smaller $7 \times 10$-dim transports conditioned on the same encoded context. What patching removes is explicit cross-patch output-covariance entries—not access to long-range information. The shared encoder routes $x$-information into every patch, so long-range structure is carried by the conditioning network rather than by a full output-side covariance, at far lower cost.

**Spectral parameterization**    Any SPD matrix admits an eigen-decomposition $U\Lambda U^\top$ where $U$ is the orthogonal matrix formed by the eigenvectors and $\Lambda$ is a diagonal matrix with nonnegative eigenvalues. Instead of first learning a dense Jacobian and then computing its spectrum as a costly after-the-fact byproduct, the model predicts the spectral factors themselves: translations, nonnegative eigenvalues, and an orthogonal basis. When we *parameterize the forecast map through its spectral factors natively*, we replace the expensive search over arbitrary matrix-valued maps with a search within the structured cone of SPD maps.

We generate *eigenvalues* $\Lambda \succeq 0$ and translation $t$ via MLP; both are $n$-dim vectors, each costing $O(n)$. The orthogonal factor $U$ is parameterized as a product of $R$ Householder reflections $H_i = I - 2v_i v_i^\top$ with $\|v_i\|_2 = 1$, so $U = H_R \cdots H_1$ (Householder, 1958). This costs $O(Rn)$, is orthogonal by construction ($U^\top U = I$), and recovers *any* rotation at $R = n$; smaller $R$ gives a reduced-capacity but cheaper alternative. The full spectral construction thus reduces from the $O(n^3)$ cost of post-hoc eigendecomposition of a dense Jacobian to $O(Rn)$. A further optimization: the involutory property of Householder reflections ($H_i^2 = I$) further enables a Triton kernel that recomputes reflections during the backward pass, yielding $O(p)$ activation memory per patch regardless of $R$ (App. A.2.2).

Parameterizing this way turns the optimal map into a sequence of *geometrically meaningful* actions: translate, rotate, scale, rotate-back, where rotate is a series of learned reflections and rotate-back is applying those in reverse. Since the Gaussian distribution is closed under affine maps, a Gaussian ball is transformed into a *Gaussian ellipsoid*. The shift can equivalently be applied before the rotation, $y_0 \mapsto U\Lambda U^\top(y_0 + t) = \tilde{\mu} + A\, y_0$ with $\tilde{\mu} = At$, which lets the network learn the "center" $t$ in the unscaled noise space.

**Bidirectional Encoder**    In practice, $x$ is first encoded into a lower-dimensional Gaussian latent $z$ via a bidirectional affine coupling network inspired by ANF (Huang et al., 2020). The SPD transport parameters $(t_y, \Lambda, U)$ are produced from the final latent representation rather than directly from the raw context window. Thus the notations $q_\theta(y \mid x)$ and $G_\theta(\cdot; x)$ in the preceding equations refers to

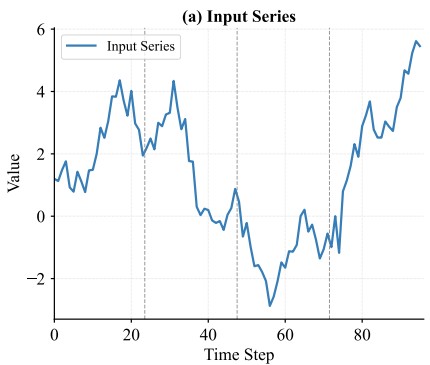
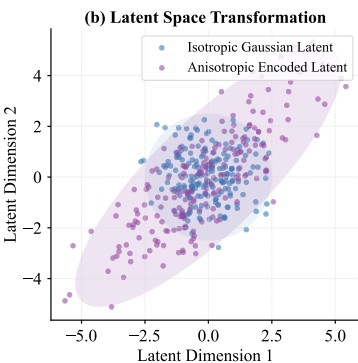
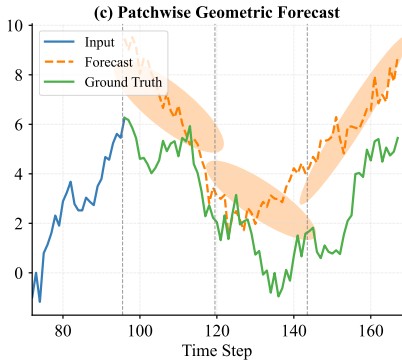

*Figure 2.* Fern forecasting mechanism. (a) A context window is processed by the bidirectional encoder. (b) Latent noise $z \sim \mathcal{N}(0, I)$ is conditionally scaled and shifted into an anisotropic Gaussian ellipsoid. (c) Output-space noise $y_0 \sim \mathcal{N}(0, I)$ is reshaped by the SPD transport into Gaussian forecast-patch ellipsoids.

the composition

$$x \longmapsto z \longmapsto (t_y, \Lambda, U) \longmapsto G_\theta(\cdot; z).$$

A fresh $z^1 \sim \mathcal{N}(0, I)$ is drawn on every forward pass. At encoder layer $i$, the current context state $x^i$ first produces features $h_x^i = H_x(x^i)$, whose head outputs the scale and shift $(s_z^i, t_z^i)$ for the latent update

$$z^{i+1} = s_z^i \odot z^i + t_z^i.$$

The updated latent state then produces features $h_z^i = H_z(z^{i+1})$, whose head outputs $(s_x^i, t_x^i)$ for the context update

$$x^{i+1} = s_x^i \odot x^i + t_x^i.$$

These affine coupling steps are invertible when scales are nonzero, so they reshape the isotropic latent into an anisotropic Gaussian while preserving the qualitative structure carried by the time-delay embedding. The resulting latent $z$ carries context information from $x$, while the separate source $y_0 \sim \mathcal{N}(0, I)$ is the forecast-space noise transported into $\hat{y}$ by the final SPD map (Alg. 1).

## 3. Evaluation

**Rethinking LTSF**   A fundamental tension in current LTSF is the **'channel independence (CI) vs channel dependent (CD) Paradox.'** A common belief we called **Information Monotonicity Assumption** clearly favors CD: it believes adding correlated variables *monotonically* increases information content, and the (*implicit oracle*) deep learner will automatically separate signal from noise in finite data, and *monotonically* improves performance as we throw more data at it. Yet, benchmarks (Nie et al., 2023; Zeng et al., 2023) show that CI models that reference only a variable's own past history *can* outperform CD models. Unless carefully designed as in (Zhang & Yan, 2023), naive inter-variable mixing hampers performance. We argue: **this is not an**

---

**Algorithm 1** Fern

**Require:** Windowed input $x$
1: $x^1 \leftarrow x$
2: $z^1 \sim \mathcal{N}(0, I); \quad y_0 \sim \mathcal{N}(0, I)$    *// fresh latent noise and forecast-space source noise*
3: **for** $i = 1$ **to** $K_{\text{enc}} = 5$ **do**
4:    $h_x^i \leftarrow H_x(x^i)$        *// encode current context state*
5:    $(s_z^i, t_z^i) \leftarrow \phi_x^i(h_x^i)$     *// context head outputs latent scale/shift*
6:    $z^{i+1} \leftarrow s_z^i \odot z^i + t_z^i$     *// affine update of latent z*
7:    $h_z^i \leftarrow H_z(z^{i+1})$      *// encode updated latent state*
8:    $(s_x^i, t_x^i) \leftarrow \phi_z^i(h_z^i)$     *// latent head outputs context scale/shift*
9:    $x^{i+1} \leftarrow s_x^i \odot x^i + t_x^i$ *// affine update of context state*
10: **end for**
11: $h_z \leftarrow H_z(z^{K_{\text{enc}}+1})$           *// final latent feature*
12: $(\Lambda, t_y, U) \leftarrow \psi(h_z)$     *// SPD head: eigenvalues, source-space shift, orthogonal factor*
13: $\hat{y} \leftarrow U \Lambda U^\top (y_0 + t_y)$ *// SPD forecast-space transport*
14: **return** $\hat{y}$

---

**architectural quirk, but a consequence of dynamical systems theory**.

By *Takens' embedding theorem* (Takens, 1981) and its stochastic extensions (Stark et al., 1999; Stark, 2003), the time-delay embedding of a *single* observable is sufficient to *reconstruct the full system's attractor* (up to diffeomorphism). The history of *one variable* $[x_{t-L}, \ldots, x_t]$ **encodes** the coupling of the system's *state variables*. When the intrinsic dimension is low, even a short patch contains *causal* system information that is topologically accurate, meaning the embedded manifold is a stretched or compressed but not torn or punctured version of the original. Concretely, $x, y, z$ of the Lorenz-63 system *interact with each other every step*. Yet, each coordinate shown previously in Fig. 1 is predicted conditionally only on the channel's own context. This shows

explicit cross-channel mixing *may not be mandatory*.

**Mori-Zwanzig formalism (Chorin et al., 2002)** helps explain why CI *can* outperform CD: it decomposes dynamics into: (1) $\mathbf{A}(t)$ the *resolved states* (part of the full state that we choose to model) chosen at time $t$ (2) $\mathbf{K}(s)$ is the *memory kernel*, i.e the systematic, history-dependent influence of all the *unresolved variables* at time $s$ on the resolved ones (3) residual *unresolved variables'* influences $\mathbf{F}(t)$ orthogonal to $\mathbf{A}(t)$. Then, $\frac{d}{dt}\mathbf{A}(t) = \mathbf{\Omega}\mathbf{A}(t) + \int_0^t \mathbf{K}(s)\mathbf{A}(t-s)\,ds + \mathbf{F}(t)$, where three terms represent Markovian terms (now), memory (past), and noise-like exogenous forcing. What CI models offer is what Takens' theorem offers: the *topologically faithful* representation of the manifold formed by *all three causal terms*, without *identifying* $\mathbf{A}(t)$, $\mathbf{K}(s)$ and $F(t)$. Where naive CD models fail is to carelessly **dilute the resolved manifold** with terms that should be in $F(t)$, e.g. treating stochastic exogenous factor such as raining or macroscopic environment, or even pure noise (spurious correlation) as microscopic innate dimension to model.

Take Lorenz-63 for an example. Single channel 100-dim TDE of $x$ has a 3-dim embedded structure inside it. Cross-channel sharing with 100-dim $y$ **in the form** $[x, y]$ offers zero additional information as it contains the same 3-dim structure up to diffeomorphism. Sharing **in the mixer style** where an MLP process $[x_i, y_i]$ can destroy both representation. Sharing with an additional noise channel $e$ further harms manifold integrity by injecting non-attractor variability. In the absence of strong prior knowledge about which channels are truly informative, CI remains a conservative choice that preserves a meaningful manifold representation.

**The Ontological Coherence of Benchmarks** Mori-Zwanzig formalism sheds light on where LTSF practices overextend the already shaky *Information Monotonicity Assumption*: (1) Benchmarks consist of loosely linked correlated variables, often little more than a collection of environment sensors. CD models default to mixing all channels—and when outperformed by CI, often attribute failure to *how* channels were blended, not *whether* they should be. When each channel is further assumed relevant *a priori*, we risk equating representation learning with feature hoarding and drowning the resolved states $\mathbf{A}(t)$ in high-dimensional noise. (2) Evaluation defaults to *all* correlated variables, and manifold-informed selection is dismissed as cherry-picking. Assuming *pointwise predictability* encourages competition on near-random walk datasets like `Exchange`, where pointwise forecasts are meaningless (see (Saqur et al., 2026)).

This issue is not unique to neural forecasting. The vector autoregression (VAR) model in econometrics also predicts a cluster of correlated variables' future based on every feature's past. Yet variables enter a VAR only when they plausibly belong to one interacting system: we know inflation, money supply, and unemployment rate are *causally intertwined* within *one economic system*. The same principle holds for physical dynamical systems such as Lorenz-63, where coordinates lie on one attractor, or for RGB channels in vision, where channels describe one physical scene. Cross-channel modeling is therefore not inherently problematic; the question is whether the channels form a coherent state description for the forecasting task.

Consider `Weather`: local air parcel thermodynamics (dew point, vapor pressure, humidity) are state variables of one system, related by Clausius–Clapeyron and ideal gas laws. Rainfall, by contrast, is sparse, heavy-tailed, and event-driven; it is causal for many outcomes, but it need not lie on the same local manifold as air-parcel thermodynamics, as it is determined by cloud physics kilometers above, not air parcel conditions. In M-Z terms, the benchmark folds $\mathbf{F}(t)$ into $\mathbf{A}(t)$. Aggregating error across all channels conflates tracking air-parcel dynamics with guessing event counts. Consider `Traffic`: hundreds of freeway sensors are spatial-temporal observables that should have been grouped into a physically meaningful vector-valued time-delay embedding; instead, their physical identities are *anonymized*, artificially turning a structured spatio-temporal layout into a generic "scale-up" challenge. CI models must treat each unlabelled sensor as isolated; CD models must pretend all 862 sensors are mutually relevant. Neither is coherent, and we are benchmarking a CD model's ability to *reject spurious cross-channel mixing*. These examples illustrate that benchmark provenance alone does not determine whether channel mixing is scientifically meaningful.

Data artifacts from poorly sanitized real-world data independently harm benchmarking. Consider `ETT`: (1) sentinel values (stuck sensors with exactly the same values) are *everywhere*: `ETTh1` has 140 entries of 8.28, 100 entries of 1.76, 120 entries of 1.73; `ETTh2` has 250 entries of -31.46 for a week and *10k entries of 88.29 that persist for months*. (2) zero-inflation: two columns have 22–33% zeroes and most zeroes are not isolated but span multiple days (Table 4). These zeroes are hard to impute, and problematic when multiple training windows contain them. For example, a sequence $[100, 101, 102, 103, 0, 105]$ (with 16% zeroes) has mean $\approx 85$ which punishes any sensible prediction of $104$ in all other windows containing this zero.

Fig. 3 illustrates this issue on the `ETTh2` HULL column. In regions dominated by dense zeros (steps 1000–1500), predictions *invent* a downtrend to reduce the pointwise error caused by zeros, and critically miss the *real upward trend* obvious to human eyes. In regions with a large amount of intermittent zeroes (prior to step 1000), predictions are consistently lower to be *metric-correct*. Models with a moving average component benefit: they see $[100, 101, 102, 103, 0, 105]$ *as* a sequence of 85's; other

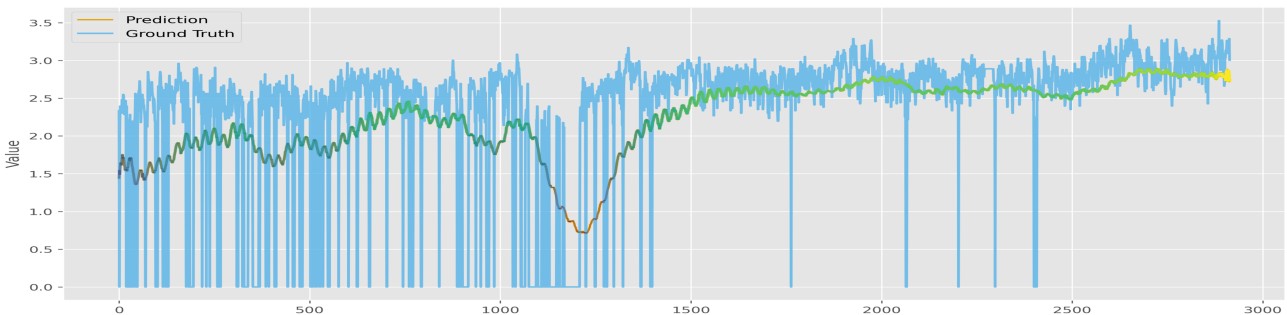

*Figure 3.* Reconstruction of ETTh2's HULL column: multiple consecutive zeros severely plague the dataset

signal-sensitive models suffer. In other words, we evaluate the model on the dual roles of forecaster and *predictive data janitor*.

| Model–Data | MSE@1 | Best | Ep | Δ |
|---|---|---|---|---|
| TST–ETTh1 (val) | 6.49 | 9.24 | 6 | +2.75 |
| TST–ETTh1 (test) | 6.85 | 6.21 | 6 | -0.64 |
| TimeMixer–ETTh1 (val) | 6.28 | 9.51 | 6 | +3.23 |
| TimeMixer–ETTh1 (test) | 7.38 | 7.34 | 6 | -0.04 |
| Fern–ETTh1 (val) | 8.24 | 12.10 | 6 | +3.86 |
| Fern–ETTh1 (test) | 18.88 | 5.51 | 6 | -13.37 |
| DLinear–ETTh1 (val) | 7.12 | 6.22 | 6 | -0.90 |
| DLinear–ETTh1 (test) | 7.79 | 7.39 | 6 | -0.40 |

*Table 1.* Recency bias on ETTh1 (MSE). @1 is epoch 1, Best is the later selected epoch; $\Delta = \text{Best} - @1$.

A final benchmarking hazard in real-world datasets is where *early-stopping* decides the winner. ETT's non-stationarity—likely low-frequency trend drift—makes training's tail statistically closer to validation than to test. Table 1 showcases the pathology: PatchTST, TimeMixer and Fern *all* converge to best test MSE at epoch 6, but the validation errors are *universally* around 45% higher than at epoch one. Then (1) naive early-stopping effectively compares DLinear against three expressive models *locked at epoch 1*; (2) fixed-epoch schemes as in (Wang et al., 2024a) *invert Fern from worst to best*. Likely, expressive models try simple extrapolation first, resulting in a deceptively low initial validation error, and getting locked in there. One minimal but principled fix is a *grace period* where we skip the first few epochs before checkpointing. In this experiment, all expressive models *do* converge at epoch 6 to the best test MSE under free run, when we set the grace period to 2 or 3. Yet, we should be aware that *early stopping is an intrinsic determinant of the leaderboard for datasets with uncontrolled drift*.

**Rethinking Benchmarking** Parallel to the *Information Monotonicity Assumption* sits a second fallacy: the *Provenance-over-Structure Fallacy*. Real-world nonstationarity is often deemed too complex to simulate, while structural generators are dismissed as "toy"—without specifying

which dynamical property the real dataset contains that the generator lacks. By extension, real-world sourced datasets are treated as inherently superior—*as if provenance were a scientific property*. But provenance only tells us where the data came from; it does not tell us whether the benchmark identifies the phenomenon we claim to measure. We should not mistake where data comes from for what it can tell us.

This matters most for nonstationarity claims. A historical dataset is *one* realized path among *many* counterfactuals: winning by a few percent MSE on that single path is not the same as mastery of that *system*. Nor does it demonstrate robustness to shocks and regime shifts. We risk trading scientific rigor for pure *historical path emulation* when we accept this chain of inference: real-world dynamics are too uniquely complex to specify; benchmarks built from real data must capture that complexity; and better MSE on those benchmarks therefore constitutes progress on "nonstationarity", i.e. the unspecifiable dynamics. This conclusion is unfalsifiable by construction; worse, some real-world data are artifact-laden enough that even the empirical improvement on them is suspect.

This problem compounds further when we stack historical datasets and infer that a broad range of nonstationarity has been covered. A closer look at the canonical benchmarks suggests that many are quasi-periodic datasets of similar difficulty: extremely simple models (linear, frequency-only) rarely fail catastrophically, and many SOTA architectures explicitly encode priors such as seasonal-trend decomposition and frequency components. In other words, the community is already taking a position on the process class, even when models tuned to those benchmarks are presented as process-agnostic. As we will see, DLinear's MSE explodes on simple chaos (Rössler), and modest shocks don't merely add error—they multiply it. The correct response, in our view, is not to dismiss chaotic data as "toy" or chaos as niche—it is neither—but to consider both the value of falsifiable stress testing and ask whether the canonical benchmarks conflate a handful of easier processes with "real-world complexity," even when we sample and test multiple datasets from them.

| Dataset | fr | | tm | | tst | | dl | | kp | | mtcn | | pfnn | |
|---|---|---|---|---|---|---|---|---|---|---|---|---|---|---|
| | MSE | WD | MSE | WD | MSE | WD | MSE | WD | MSE | WD | MSE | WD | MSE | WD |
| *Standard Chaotic Dynamics* | | | | | | | | | | | | | | |
| Rossler-Base | **0.019** | **0.011** | 1.03 | 0.903 | 2.45 | 2.25 | 5.42 | 5.06 | 11.94 | 5.58 | 0.47 | 0.42 | 21.05 | 16.64 |
| Rossler-Param | **0.036** | **0.017** | 3.49 | 2.91 | 10.02 | 8.62 | 28.74 | 25.42 | 25.07 | 17.91 | 1.64 | 1.37 | 28.09 | 23.08 |
| Lorenz-Base | **21.66** | **4.41** | 43.21 | 10.09 | 38.89 | 10.56 | 76.55 | 39.34 | 95.50 | 11.05 | 26.02 | 5.96 | 198 | 120 |
| Lorenz-State | **19.26** | **3.73** | 48.81 | 10.22 | 40.71 | 10.90 | 70.36 | 35.58 | 97.85 | 13.52 | 28.49 | 5.68 | 210 | 122 |
| Lorenz-Param | **25.21** | **4.61** | 52.10 | 10.63 | 40.59 | 9.19 | 70.69 | 32.89 | 103 | 17.86 | 35.96 | 7.57 | 219 | 151 |
| Lorenz96-Base | **5.19** | **1.33** | 8.03 | 2.97 | 6.35 | 2.49 | 10.98 | 6.02 | 17.42 | 3.41 | 10.38 | 3.64 | 20.17 | 15.56 |
| Lorenz96-Switch | **9.56** | **3.14** | 11.96 | 5.34 | 10.73 | 4.73 | 13.68 | 7.80 | 21.61 | 4.59 | 11.78 | 5.25 | 24.42 | 17.90 |
| Chua-Base | 0.056 | 0.033 | 0.094 | 0.046 | 0.186 | 0.119 | 0.720 | 0.507 | 1.11 | 0.482 | **0.051** | **0.029** | 1.77 | 1.67 |
| Chua-Param | 0.021 | 0.011 | **0.013** | **0.008** | 0.097 | 0.078 | 0.681 | 0.559 | 0.944 | 0.353 | 0.030 | 0.021 | 2.31 | 2.14 |
| Chua-Switch | 0.178 | 0.106 | 0.174 | 0.123 | 0.318 | 0.230 | 0.770 | 0.591 | 1.11 | 0.584 | **0.099** | **0.068** | 1.74 | 1.64 |
| *Switching Linear Dynamical System* | | | | | | | | | | | | | | |
| SLDS-Base | 2.84 | 1.46 | 4.54 | 2.80 | 2.27 | **1.05** | 4.42 | 3.50 | 2.96 | 1.96 | **1.96** | 1.10 | 3.52 | 2.97 |
| SLDS-Param | 2.36 | 1.36 | 2.60 | 1.58 | **2.18** | **0.91** | 2.26 | 1.50 | 2.19 | 1.38 | 2.30 | 1.80 | 2.57 | 2.13 |
| SLDS-Switch | **4.05** | **2.02** | 7.84 | 4.84 | 9.56 | 5.19 | 4.73 | 3.38 | 4.56 | 3.38 | 9.47 | 5.82 | 8.23 | 6.81 |
| *Seasonal AR Shocks (SAR)* | | | | | | | | | | | | | | |
| SAR-Base | **0.055** | 0.011 | **0.055** | 0.013 | 0.074 | 0.021 | 0.056 | **0.010** | 0.056 | 0.013 | **0.055** | 0.013 | 1.85 | 1.21 |
| SAR-Param | **0.355** | **0.053** | 0.361 | 0.065 | 0.480 | 0.128 | 0.380 | **0.053** | 0.366 | 0.075 | 0.359 | 0.065 | 10.83 | 7.43 |
| *GARCH Volatility Shocks* | | | | | | | | | | | | | | |
| GARCH-Base | **0.227** | 0.220 | 0.234 | 0.201 | 0.271 | **0.174** | 0.264 | 0.190 | 0.255 | 0.217 | 0.261 | 0.210 | 0.255 | 0.208 |
| GARCH-Param | **0.177** | 0.174 | 0.186 | 0.156 | 0.220 | 0.138 | 0.183 | **0.135** | 0.191 | 0.172 | 0.206 | 0.163 | 0.236 | 0.196 |
| *Double-Well Potential (DW)* | | | | | | | | | | | | | | |
| DW-Base | 0.054 | **0.028** | 0.059 | 0.043 | 0.091 | 0.057 | 0.055 | 0.034 | **0.049** | 0.042 | **0.049** | 0.043 | 1.47 | 1.41 |
| DW-Param | **0.682** | **0.506** | 1.08 | 0.843 | 0.983 | 0.721 | 0.847 | 0.711 | 1.03 | 0.856 | 1.00 | 0.815 | 0.837 | 0.756 |
| *Ornstein–Uhlenbeck (OU) Diffusions* | | | | | | | | | | | | | | |
| OU-Base | **0.234** | 0.195 | 0.251 | 0.131 | 0.273 | **0.112** | **0.234** | 0.166 | 0.251 | 0.156 | 0.241 | 0.178 | 0.237 | 0.216 |
| OU-Param | **0.239** | 0.170 | 0.251 | 0.131 | 0.273 | **0.112** | **0.239** | 0.163 | 0.251 | 0.156 | 0.241 | 0.178 | 0.383 | 0.358 |
| *Real-World Benchmarks* | | | | | | | | | | | | | | |
| ETTm1 | 8.97 | **5.37** | 9.12 | 5.63 | **8.83** | 5.49 | 9.71 | 6.25 | 9.22 | 5.69 | 11.14 | 7.11 | 43.69 | 34.25 |
| ETTh1 | 10.97 | 5.75 | 10.51 | 5.23 | 10.97 | **4.83** | **10.39** | 5.00 | 10.75 | 5.55 | 14.52 | 9.01 | — | — |

*Table 2.* **Stress-testing with stochastic and chaotic systems and controlled non-stationarity.** Models: fr (Fern), tm (TimeMixer), tst (PatchTST), dl (DLinear), kp (Koopa), mtcn (ModernTCN), pfnn (PFNN). Lower is better. **Purple**: best; Light Purple: second-best; Light Orange: diverged; '—': catastrophic.

**Beyond Pointwise Metrics** Pointwise metrics penalize phase shifts: $[0, 1, 2, 3]$ and $[1, 2, 3, 4]$ are intuitively similar, as are $[0, 2, 4, 6]$ and $[2, 4, 6, 8]$, yet the same one-step shift yields $4\times$ larger MSE. They leave *no room for error*: correct predictions arriving slightly too early or late are treated as entirely wrong. DTW (Sakoe & Chiba, 1978) partially addresses phase but is computationally heavy and non-differentiable. We complement MSE with the *Wasserstein-2 (W2) distance*, which measures the minimal "work" required to morph one distribution into another, rewarding sharp forecasts even under phase shift. Generally this is expensive, and the computationally feasible option is *Sliced Wasserstein Distance (SWD)* that project onto $L$ random directions and then average the resulting 1D W2 distances (Bonneel et al., 2015). Luckily for CI models, we can treat each channel's forecast and target as empirical 1D distributions; W2 reduces elegantly to the squared $L_2$ distance between sorted values: $\text{WD}^2(y^\star, y) = \frac{1}{H} \sum_{h=1}^{H} \left( y_{(h)}^\star - y_{(h)} \right)^2$, where $y_{(1)}^\star \leq \cdots \leq y_{(H)}^\star$ and $y_{(1)} \leq \cdots \leq y_{(H)}$ are order statistics (Peyré & Cuturi, 2019). This makes WD *index-agnostic*: MSE asks whether the value is correct at the exact time index, while WD asks whether the forecast produced the right set of values over the horizon. Our 1D-W2 proposal converges with ideas independently developed in several fields (Muskulus & Verduyn Lunel, 2011; Wiesel, 2022; Aoun et al., 2024).

**Effective Prediction Time (EPT)** measures the first forecast step at which absolute error exceeds one training-set standard deviation:

$$\text{EPT}_{b,d} = \min\{s \in \{1, \dots, H\} : |y_{b,d,s}^{\text{pred}} - y_{b,d,s}^{\text{true}}| > \epsilon_d\},$$

with $\text{EPT}_{b,d} = H$ if the threshold is never exceeded; we report the mean $\text{EPT}_{\text{avg}}$ across batches and channels. EPT turns a long-horizon error curve into a reliability horizon: if multiple models reach $\text{EPT} \approx 190$ on a 336-step task, steps $t > 190$ should be treated as failure-mode analysis rather than actionable signal.

**The Gold-Standard: Synthetic Benchmarking** If, with Arnold, *mathematics is the part of physics where experi-*

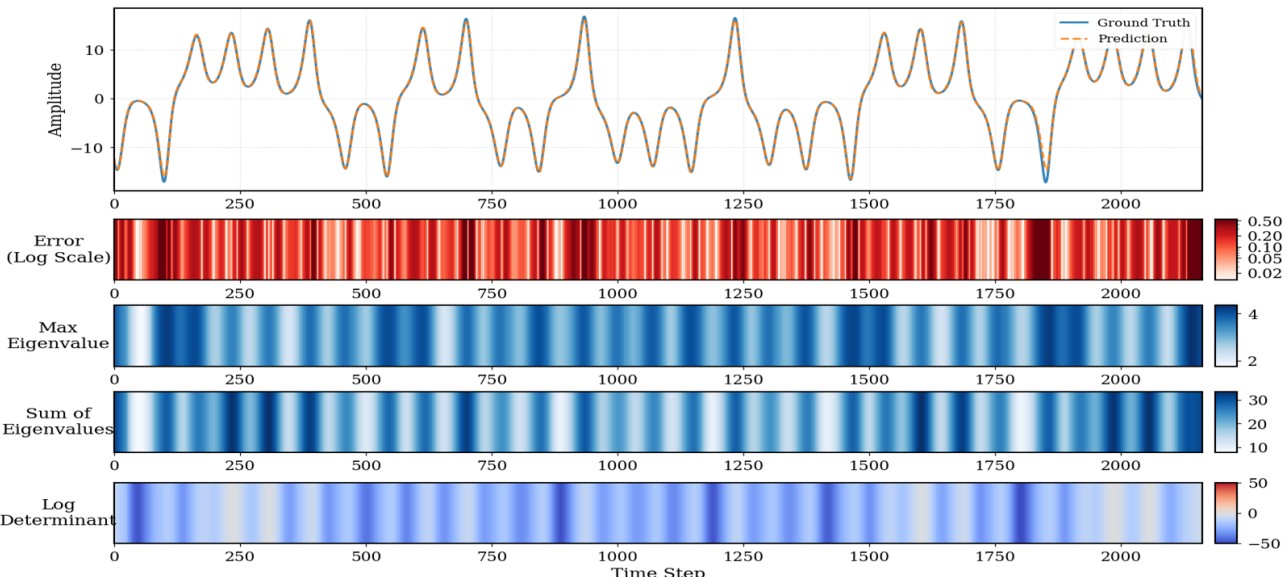

*Figure 4.* Diagnostics for Lorenz-63 x. Top to bottom: forecast (orange) vs. ground truth (blue); log absolute error; mean patch-wise max eigenvalue; mean patch-wise sum of eigenvalues; mean patch-wise log of products of eigenvalues.

*ments are cheap*, then *synthetic data is forecasting where the ground truth is actually true*. By picking systems with attractors and structured invariant measures, *many problems with real-world datasets vanish*: (1) no measurement errors and data-quality artifacts; (2) distributional properties are analytical, not assumed or inferred; (3) random walk and non-dissipative systems are ruled out, risks of measuring another `Exchange` avoided; (4) nonstationary shock is defined by the user; recency bias between validation and train is eliminated unless deliberately allowed. This also separates two failure modes that real-world benchmarks can conflate: true negatives, models that cannot learn even clean generated dynamics, and false positives, models that exploit historical artifacts but fail on clean data.

We design three controlled nonstationary scenarios, *all occurring exactly at the midpoint of the training trajectory*: (1) **Parameter drift:** system parameters shift slightly (with same initial conditions); (2) **State perturbation:** state variables receive an additive shock (with same parameters); (3) **Regime replacement:** the trajectory switches to a different one (different parameters *and* initial conditions). Exact numerical parameters are reported in App. Table 7.

In our view, *synthetic benchmarks are software tests for forecasters*—refusing them is like skipping sensible edge-case tests in favour of a decades-old test suite from a different domain and calling it your 'real-world' CI/CD. Dismissing these controlled environments as "not real" is akin to dismissing wind tunnel air as "not real wind". We are not against well-curated real-world data: we only showcase that the key to *falsifiable claims* is to make the *type, timing, magnitude, and before/after distributions* of shocks all explicit

and precise. In fact, nonstationarity is rarely mystical for industry practitioners: many can *articulate* what makes their domain's nonstationarity special, and in controllable settings (robotics, factory floors) experts can sometimes intervene on the process directly—designing controlled shocks rather than only describing them. Inviting domain experts to specify likely shock scenarios and then testing model stability under those scenarios is far stronger than hoping `ETT` covers industry-specific nonstationarity.

Table 2 reports the shock-table evaluation. We compare Fern against five LTSF baselines and one invariant-measure learner: DLinear (Zeng et al., 2023), TimeMixer(Wang et al., 2024a), PatchTST(Nie et al., 2023), Koopa (Liu et al., 2023), ModernTCN (Luo & Wang, 2024) and PFNN (Cheng et al., 2025) on 21 synthetic dynamical systems and 2 ETTs using 336-step input and 336-step forecast horizons, averaging results over 2 random seeds. Table 9 presents tests with 4 random seeds, 4 prediction horizons (96, 192, 336, 720) averaged results on Fern and TimeMixer, PatchTST and DLinear with standard errors on MSE. We use a 3-epoch grace period in Table 2, while Table 9 reports the raw no-grace-period protocol, to show sensitivity to checkpointing choices. Settings in Appendix A.4, Appendix A.2.1.

Table 2 shows that the *diffusively blurred geometry* of stochastic systems poses fewer difficulties than chaotic dynamics and nonstationary shocks, where the challenge is to locate the relevant portion of a sharp but sensitive manifold. Simple models such as DLinear and Koopa rank 1st or 2nd on some stochastic systems and on the less problematic ETT1. Yet on Rössler, one of the easier chaotic systems, DLinear and Koopa achieve MSE of 5.42 and

| Variant | H1 MSE↓ | L63 MSE↓ |
|---|---|---|
| Base | 10.96 | **21.66** |
| No enc. + no $\mu$ upd. | n/a | 194.43 |
| Only encoder | 11.17 | 27.09 |
| No rotation | 11.84 | 27.62 |
| No patch | 10.99 | 22.86 |
| Reflections = 2 | 11.52 | 26.14 |
| Reflections = 24 (8-block) | **10.91** | 23.92 |

*Table 3.* **Ablations (MSE) on ETTh1 & Lorenz-63** 336-in-336-out. Full definitions and metrics are reported in App. 8

11.94 respectively, versus Fern's 0.019—errors $285\times$ and $628\times$ larger—and the gap widens to $790\times$ and $696\times$ under parameter shift. This contrast is precisely what the stress test is designed to expose: average performance on quasi-periodic or weakly stochastic benchmarks can hide failure modes that appear immediately under controlled chaotic dynamics. Forecasting is ultimately an aid to decision making: worst-case robustness matters as much as mean matching.

By contrast, Fern is strongest on the chaotic and nonstationary stress tests—including a 98% MSE reduction relative to TimeMixer on Rössler in the base setting—and remains competitive across most synthetic settings. On Lorenz-63, baselines collapse to mean-guessing early—DLinear at horizon 96, TimeMixer at 192, PatchTST at 336 (Table 15)—while Fern maintains pointwise accuracy until horizon 720. At 720 steps (roughly 6.5 Lyapunov times, where errors amplify $\approx 650\times$ beyond useful pointwise precision), **geometry persists**: SWD 4.89 versus 10–40 for baselines.

Fig. 4 shows the *eigen-profile* of Fern. When MSE is low ($1/8$ of standard deviation), it reveals additional insights: (1) *spectral radius* spikes at violent lobe switches, coinciding with the largest errors—the model 'believes' these segments are hardest and gets validation; (2) *trace* spikes intra-lobe, suggesting less dominant eigen-directions activate during stable traversal; (3) *log determinant* captures volume scaling. Notably, some lowest-error regions (pale, row 2) coincide with low trace and strongly negative log-determinant immediately before error spikes. This suggests a specific failure mode: the model becomes overconfident in its mean prediction, collapses variance, and consequently suffers when the system transitions unexpectedly.

Table 3 is a simple *ablation summary* with full details in Table 8. We find: (i) Bidirectional encoder successfully compresses the input information into the latent space; without encoder, the model is useless, MSE and SWD explode; (ii) In the 'only encoder' case, we disable the repeated translation in the decoder. This hurts ETTh1 and, more strongly, Lorenz, suggesting that the data-dependent translations are particularly important for chaotic systems; (iii) removing either the eigen-based rotation or the patching mechanism consistently degrades Lorenz performance and

often harms ETTh1, confirming that local geometric alignment and patch-wise conditioning are both doing real work; (iv) Increasing reflection capacity improves ETTh1 in this grid but doesn't uniformly improve Lorenz-63, where R=8 remains strongest. This suggests the $R$ acts as both capacity and regularization; low-dimensional attractor structure can make small R sufficient, while noisy or higher-dimensional settings may benefit from additional reflections.

## 4. Literature Review and Discussion

The LTSF field has been dominated by Transformer variants (Zhang & Yan, 2023; Liu et al., 2024). Foundational critiques (Zeng et al., 2023; Saqur et al., 2026) on model complexity and evaluation paradigms catalyzed a shift to simplicity and interpretability. Recent efforts emphasize linearity and efficiency (Xu et al., 2024b; Huang et al., 2025), and frequency-domain analysis for periodic signals (Wang et al., 2025; Wu et al., 2023). **Koopman operator theory** (Brunton et al., 2022), is another direction with physics-informed design priors (Liu et al., 2023; Zhang et al., 2025). Emerging focus on endogenous/exogenous structure (Qiu et al., 2026; Wang et al., 2024b) also provides a complementary angle to our discussion. Our ellipsoidal method can be extended to conformal prediction like (Xu et al., 2024a) in future work. *Crucially*, our *conditional* task *differs from learning **global** invariants* (e.g., Koopman operators or Neural ODEs); we clarify scope in App. A.1.1 and show one global learner's catastrophic divergence on conditional tasks in main text, with discussion in App. A.1.2.

## 5. Limitation and Future Work

First, though not explored in the paper, $y_0 \sim \nu = \mathcal{N}(0, I)$ can be relaxed: if the context window $x$ and target $y$ share the same dim, then using $\mathcal{N}(x, I)$ as reference allows us to directly probe the dynamics; we take $\nu$ to be Gaussian to simplify the math and restrict the forecast into Gaussian. *Key structural advantage*: between any two Gaussians, the unique W2 OT map is *strictly affine*, meaning W2-optimal map is analytical and simple and not the linearization of some *implicit* non-linear map. However, Brenier's theorem is quite general and we *can* swap Gaussian for other classes if "Gaussian prediction" feels too restricted. Second, this approach spans both pointwise prediction *and* probabilistic prediction since model outputs $\Sigma_\theta$ directly. We focus on point forecasts, and leave a full probabilistic evaluation (e.g., NLL/CRPS) to future work.

**Code availability.** The implementation, training logs, and synthetic benchmark generators (21 dynamical systems with controllable shock scenarios) are available at `https://github.com/QilinWang/Fern-icml2026`. A standalone release of the benchmark generators is planned.

## Acknowledgements

The author dedicates this work to his father, who passed away in 2023. His curiosity about prediction, and the hope of writing something he would have loved to read, were a deep motivation for entering this field and enduring the long process behind this paper. I wish he could have seen it completed.

## Impact Statement

This paper presents work whose goal is to advance the field of Machine Learning. There are many potential societal consequences of this work, none that the author feels must be specifically highlighted here.

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

# A. Appendix

## A.1. App: Scope and Additional Discussions

### A.1.1. FERN'S ROLE

**What Fern is, and what Fern is not**  We clarify the scope of Fern relative to chaos-specialized simulators, governing-equation learners, and invariant-measure learners. Fern is *not* an iterated one-step chaos specialist. Such models recursively feed their own output to simulate long trajectories, and chaos-tuned variants (e.g., DSDL) excel on Lorenz-63 (Wang & Li, 2024). Fern is instead a direct, multi-step forecaster for general-purpose tasks, stress-tested on chaotic datasets only as a supplement to standard LTSF benchmarks. The metric EPT, which we advocate for, is a common measure of one-step *model stability* of chaotic dynamics, alongside variants like VPT (valid prediction time). We use EPT to measure *dataset predictability*. Direct comparison between iterated and multi-step models can be misleading, as multi-step models are not optimized for iterated rollout.

Fern is *not* a governing-equation or invariant-measure learner. It aims at discrete-time conditional forecasting with data-driven spectral factors rather than system identification, distinct from (i) sparse governing equation finding SINDy/HAVOK/VINDy (Brunton et al., 2016; 2017; Conti et al., 2024) (ii) Neural-ODE identification (Ko et al., 2023) (iii) invariant-measure learner (Jiang et al., 2023; Schiff et al., 2024).

### A.1.2. MARKOV NEURAL OPERATORS

Operator-learning approaches (PFNN, MNO (Li et al., 2022)) are *invariant measure* learners, which pursue a complementary agenda: learning global evolution operators for specific systems. In contrast, Fern targets local conditional geometry for generic forecasting tasks, prioritizing per-window spectral diagnostics over explicit operator recovery.

However, many requests insist on probing Fern's pointwise performances vs this class, and so we complied. In the main text, Table 2 suggests PFNN performs reasonably on some smooth diffusion tasks (OU, DW), but often exhibits very large errors or outright divergence on chaotic and heavily forced settings (e.g., Rössler, Lorenz, ETTm1), under the unified training protocol. To assess robustness to data artifacts, we test on the ETTh2 case, and the reported error reaches $\mathcal{O}(10^9)$.

We were initially concerned about the errors and performed additional hyperparameter (learning rate, latent dimension size, network architecture etc) searches. Our implementation carefully follows the PFNN public repository. Based on (Cheng et al., 2025), the reported NRMSE is $0.49$ on $\approx 80,000$ samples with $\mathrm{d}t = 10^{-2}$ for Lorenz–63. Since

the standard deviation of the Lorenz system is about $8$, this corresponds to roughly 15–20 MSE over a 100-step rollout. In contrast, Fern attains an MSE of about $1.2$ on 22k samples with a 96-step prediction horizon. We conclude that the reported error on 336-step rollout is likely consistent with the paper.

**Why the difference with Markov System vs Conditional Forecasting?**  Conceptually, LTSF models take a history window of shape $[\text{batch}, \text{feature}, \text{seq}_{0:100}]$ and predict a future window $[\text{batch}, \text{feature}, \text{seq}_{100:200}]$, often with *channel-independent* parametrization, whereas Markov(-type) operators act along the temporal axis, mapping $[\text{batch}, \text{seq}_{0:1}, \text{feature}]$ to $[\text{batch}, \text{seq}_{1:2}, \text{feature}]$ by evolving the state one step at a time using *only* the *current state* information.

The task of MNO is to learn a single time-stepping operator $T$ that maps any current state to its successor; this operator is fixed and must be globally correct across all of state space, making MNO a form of *system identification*. Fern, by contrast, learns a family of *conditional* distributions $p_\theta(y \mid x)$ whose parameters vary with the context window $x$—a different distribution for each input. A globally correct operator must hold everywhere in state space; a conditional forecast distribution needs only local accuracy near observed contexts. The former is strictly harder. Indeed, while Lorenz and similar systems are Markov systems, empirically past information helps localize the system on its attractor: Lorenz-63 has 3 state variables, roughly 10 delay coordinates are enough to invoke Takens' Embedding Theorem, so the remaining input length serves as historical context for Fern's specialized local operator. We view this not as a flaw in PFNN per se, but as evidence that architectures tuned for Markovian operator learning do not transfer automatically to windowed multi-horizon forecasting; conversely, Fern does not currently target PDE operators and should not be read as a replacement in that regime.

## A.2. Additional Discussion

**Zero-inflation analysis.**  Tables 4, 5, and 6 detail zero patterns. ETTh2 and Weather are heavily affected; ETTh1 is better behaved. Traffic and Electricity show similar issues (analysis omitted for space).

**Preprocessing policy.**  ETTh2 and ETTm2 are not salvageable via imputation. Our policy is a preliminary proposal and we leave refinement to future work:

- Sentinel values $> 10\%$ or zeros $> 15\%$: drop column.

- Zeros 10–15%: apply $\mathrm{asinh}$ transform.

- Longest zero run $> 1$ week: drop column.

- Longest zero run $> 3$ hours: delete affected rows.

- Longest zero run $\leq$ 3 hours: forward–backward fill.

We apply this *only* to ETTh1/m1 in the shock tables (see App. A.4). These datasets are well-behaved, so *no columns are dropped and only zero imputation and row removals are performed.*

| Column | Total Zeros | % Zeros | Isolated | Clustered |
|---|---|---|---|---|
| HUFL | 58 | 0.33% | 1 | 57 |
| HULL | 3836 | 22.02% | 163 | 3673 |
| MUFL | 0 | 0.00% | 0 | 0 |
| MULL | 1067 | 6.13% | 245 | 822 |
| LUFL | 1188 | 6.82% | 184 | 1004 |
| LULL | 5792 | 33.25% | 414 | 5378 |
| OT | 1 | 0.01% | 1 | 0 |

*Table 4.* Zero–inflation patterns in `ETTh2`. "Clustered" counts zeros that belong to runs of length $> 1$ (consecutive zeros)

| Column | Total Zeros | % Zeros | Isolated | Clustered |
|---|---|---|---|---|
| HUFL | 89 | 0.51% | 31 | 58 |
| HULL | 410 | 2.35% | 256 | 154 |
| MUFL | 97 | 0.56% | 19 | 78 |
| MULL | 236 | 1.35% | 137 | 99 |
| LUFL | 60 | 0.34% | 1 | 59 |
| LULL | 212 | 1.22% | 7 | 205 |
| OT | 111 | 0.64% | 28 | 83 |

*Table 5.* Zero–inflation patterns in `ETTh1`. "Clustered" counts zeros that belong to runs of length $> 1$ (consecutive zeros).

### A.2.1. CHECKPOINT STRATEGY

Table 1 (using an earlier Fern variant) demonstrates that early validation error is often *anti-predictive* of final test error, rendering it an *unreliable proxy* for generalization. Naive early stopping tends to trap expressive models in local minima typical of Epochs 1–2.

To make this choice explicit, we use the following protocol:

- **Shock table.** We use a uniform 3-epoch grace period for all models before checkpoint selection.

- **Detailed horizon tables.** We do not use this grace period, preserving the raw checkpointing behavior for comparison.

- **Smoothing.** We do not use exponential smoothing of the validation objective anywhere in the paper.

### A.2.2. COMPUTATIONAL AND MEMORY COMPLEXITY

Let $n$ be the forecast horizon dimension, $p$ the patch dimension, $n_p = n/p$ the number of patches, $B$ the batch size, $d_h$ the hidden width of the coupling MLPs, $K_{\text{enc}}$ the number of encoder layers, and $R$ the number of Householder reflections per patch.

| Column | Total Zeros | % Zeros | Isolated | Clustered |
|---|---|---|---|---|
| p (mbar) | 0 | 0.00% | 0 | 0 |
| T (degC) | 6 | 0.01% | 4 | 2 |
| Tpot (K) | 0 | 0.00% | 0 | 0 |
| Tdew (degC) | 39 | 0.07% | 39 | 0 |
| rh (%) | 0 | 0.00% | 0 | 0 |
| VPmax (mbar) | 0 | 0.00% | 0 | 0 |
| VPact (mbar) | 0 | 0.00% | 0 | 0 |
| VPdef (mbar) | 3026 | 5.74% | 45 | 2981 |
| sh (g/kg) | 0 | 0.00% | 0 | 0 |
| H2OC (mmol/mol) | 0 | 0.00% | 0 | 0 |
| rho (g/m$^3$) | 0 | 0.00% | 0 | 0 |
| wv (m/s) | 0 | 0.00% | 0 | 0 |
| max. wv (m/s) | 1 | 0.00% | 1 | 0 |
| wd (deg) | 1 | 0.00% | 1 | 0 |
| rain (mm) | 50785 | 96.37% | 177 | 50608 |
| raining (s) | 49316 | 93.59% | 115 | 49201 |
| SWDR (W/m$^2$) | 24749 | 46.97% | 21 | 24728 |
| PAR ($\mu$mol/m$^2$/s) | 24614 | 46.71% | 0 | 24614 |
| max. PAR ($\mu$mol/m$^2$/s) | 24209 | 45.94% | 0 | 24209 |
| Tlog (degC) | 0 | 0.00% | 0 | 0 |
| OT | 0 | 0.00% | 0 | 0 |

*Table 6.* Zero–inflation patterns in `Weather`. "Clustered" counts zeros that belong to runs of length $> 1$ (consecutive zeros).

**Time (FLOPs)** Each encoder layer applies bidirectional affine couplings per patch. Up to fixed MLP-depth constants, the feature maps and scale/shift heads cost $O(pd_h)$ per patch. Across $K_{\text{enc}}$ layers, $n_p$ patches, and batch size $B$, the encoder cost is

$$O\big(B\, n_p\, K_{\text{enc}}\, pd_h\big).$$

The SPD head $\psi$ outputs $p$ eigenvalues, a shift $t_y \in \mathbb{R}^p$, and $R$ Householder vectors in $\mathbb{R}^p$. Thus the output dimension of the head is $(R+2)p$, giving head-generation cost

$$O\big(B\, n_p\, (R+2)\, pd_h\big).$$

Applying the $R$ Householder reflections costs $O(B\, n_p\, Rp)$. Therefore,

$$\mathcal{T}_{\text{Fern}} = O\big(B\, n_p[K_{\text{enc}}pd_h + (R+2)pd_h + Rp]\big).$$

Equivalently, since $n = n_p p$,

$$\mathcal{T}_{\text{Fern}} = O\big(B\, n[(K_{\text{enc}} + R + 2)d_h + R]\big),$$

which is linear in the horizon length $n$ for fixed patch size, hidden width, encoder depth, and reflection count.

**Memory** During training, the dominant memory cost comes from storing activations across encoder layers and patches. We store intermediate states $x^i, z^i \in \mathbb{R}^p$ and features $h_x^i, h_z^i$ of width $d_h$ for each of the $K_{\text{enc}}$ layers, yielding

$$O\big(B\, n_p\, K_{\text{enc}}\, \max(p, d_h)\big)$$

activation memory up to constant factors. We never materialize a dense $n \times n$ SPD matrix: the map is kept in factored Householder–diagonal–Householder form, avoiding the $O(n^2)$ memory footprint of a full covariance.

The eigenvalues and shift add $O(B\,n_p\,p)$ storage. The predicted Householder vectors add $O(B\,n_p\,R\,p)$. If the intermediate outputs of the $R$ sequential reflections are stored naively during backpropagation, they add another $O(B\,n_p\,R\,p)$ activation term. The Triton kernel described below instead recomputes reflections during the backward pass, reducing this intermediate-reflection activation term to $O(B\,n_p\,p)$ at the cost of extra reflection applications.

**Memory-friendly Householder reflections.** Each Householder reflection $H_r = I - 2v_r v_r^\top$ is *involutory*: $H_r^2 = I$, so $H_r^{-1} = H_r$. A naive forward pass stores all $R$ intermediate states $y^{(0)}, y^{(1)}, \ldots, y^{(R)}$ (where $y^{(r)} = H_r y^{(r-1)}$) for the backward pass, giving $O(Rp)$ activation memory per patch. The involution property lets us store only the final $y^{(R)}$ and recover earlier states during backward by applying reflections in reverse: $y^{(r-1)} = H_r y^{(r)}$. Each reflection $H_r y = y - 2(v_r^\top y)\,v_r$ costs $O(p)$ FLOPs—one dot product and one fused multiply-add per element—so the recomputation adds $O(Rp)$ FLOPs in backward while activation memory drops to $O(p)$ per patch, independent of $R$. We implement this as a Triton kernel. The optimization is a memory–time trade-off analogous to gradient checkpointing; it does not change the mathematical parameterization.

### A.2.3. ARCHITECTURAL DETAILS

Eigenvalues and translation are parameterized with differentiable soft bounds for numerical stability. We use a soft-clamp that behaves linearly within $[\ell, u]$ and saturates smoothly outside. For the elementwise scaling used in SPD matrix and most bidirectional blocks, we use $s \in [0, 5.5]$ for sensible compromise between expressivity and gradient stability, which is notoriously difficult for this kind of affine coupling structure. We use block-diagonal scaling to mimic complex multiplication—interleaved on the $x$-side in 2 of 5 encoder coupling layers to capture potential rotational dynamics on the data side. For these we use $s \in [-4.5, 4.5]$. Shift vectors $t_y$ use wider bounds $[-15, 15]$.

### A.3. App: Systems and Datasets

#### A.3.1. CHAOTIC SYSTEMS

Prior work shows that deterministic finite-precision arithmetic can suppress chaos and produce spurious periodic orbits in low precision, and that chaotic systems are highly sensitive to numerical precision and discretization choices. See, e.g., Klöwer et al. (2023) on periodic orbits at low precision and mitigation via stochastic rounding; Teixeira, Reynolds & Judd (2007) on decoupling times and truncation-

error growth; and Liao (2014) on the joint impact of truncation and round-off errors on long-time chaotic simulations. (Klöwer et al., 2023; Teixeira et al., 2007; Liao & Wang, 2014). Therefore, all chaotic ODE datasets (Lorenz-63, Rössler, Chua, Lorenz96) were generated in `float64` using the 4th-order Runge-Kutta (RK4) method (Butcher, 1987) and then converted to standard `float32` Pytorch Tensor. Pilot runs observed materially larger forecast errors across all models with `float64`.

**Lorenz-63** A canonical 3-D system modeling atmospheric convection (Lorenz, 1963):

$$\dot{x} = \sigma(y - x), \quad \dot{y} = x(\rho - z) - y, \quad \dot{z} = xy - \beta z,$$

**Rössler** A 3-D system with a folded-band attractor (Rössler, 1976):

$$\dot{x} = -y - z, \quad \dot{y} = x + ay, \quad \dot{z} = b + z(x - c),$$

**Chua's Circuit** A 3-D piecewise-linear circuit model with a double-scroll attractor (Chua et al., 1986):

$$\dot{x} = \alpha\big(y - x - h(x)\big), \quad \dot{y} = x - y + z, \quad \dot{z} = -\beta\,y,$$

with

$$h(x) = m_1 x + \tfrac{1}{2}(m_0 - m_1)\big(|x + 1| - |x - 1|\big).$$

**Lorenz96.** A $d$-dimensional toy model for mid-latitude atmospheric dynamics (Lorenz, 1996), with cyclic nearest-neighbour coupling and constant forcing forcing $= F$ (parameter `forcing` in code, dimension `dim`):

$$\dot{x}_j = \big(x_{j+1} - x_{j-2}\big)\,x_{j-1} - x_j + F, \qquad j = 1, \ldots, d,$$

with indices taken modulo $d$ (e.g. $x_0 = x_d$, $x_{-1} = x_{d-1}$).

#### A.3.2. STOCHASTIC SYSTEM

**Ornstein–Uhlenbeck (OU)** A 1-D mean-reverting diffusion with linear drift towards a long-run mean $\mu$ and Gaussian noise, with parameters $(\theta, \mu, \sigma)$ (`theta`, `mu`, `sigma` in code):

$$dX_t = \theta\big(\mu - X_t\big)\,dt + \sigma\,dW_t,$$

integrated with an Euler–Maruyama scheme using step size `dt`.

**Double-well SDE** A 1-D bistable diffusion in a double-well potential, parameterized by a shape parameter $a$ and noise scale $\sigma$ (`a`, `sigma` in code):

$$dX_t = \big(aX_t - X_t^3\big)\,dt + \sigma\,dW_t.$$

The deterministic drift $aX_t - X_t^3$ creates two metastable wells around $\pm\sqrt{a}$, with noise-driven transitions between wells.

**Switching linear (SLDS)** A 1-D switching linear dynamical system (SLDS) with two linear-Gaussian regimes, specified by $(A_1, Q_1)$ and $(A_2, Q_2)$ and Markov self-transition probabilities $p_{11}, p_{22}$ (A1, Q1, A2, Q2, p11, p22 in code). Let $s_t \in \{1, 2\}$ be the latent regime:

$$x_{t+1} = A_{s_t} x_t + \eta_t, \qquad \eta_t \sim \mathcal{N}(0, Q_{s_t}),$$

$$\mathbb{P}[s_{t+1} = i \mid s_t = i] = p_{ii}, \quad i \in \{1, 2\}.$$

This yields piecewise-linear dynamics with regime switches driven by a 2-state Markov chain.

**Seasonal AR** A 1-D discrete-time process with an autoregressive term and an explicit seasonal component of period $S$ (S) and slowly drifting amplitude. With AR coefficient $\phi$ (phi), innovation scale $\sigma$ (sigma), initial seasonal amplitude $a_0$ (a0), and linear drift rate amp_drift_per_step, we write

$$a_t = a_0 + t \cdot \text{amp\_drift\_per\_step},$$

$$x_t = a_t \cos\left(\frac{2\pi t}{S}\right) + \phi x_{t-1} + \sigma \varepsilon_t, \qquad \varepsilon_t \sim \mathcal{N}(0, 1).$$

This models a gradually drifting seasonal pattern on top of an AR(1) background.

**GARCH(1,1)** A discrete-time volatility process with conditionally Gaussian returns and autoregressive conditional variance, parameterized by $(\omega, \alpha, \beta)$ (omega, alpha, beta in code):

$$x_t = \sigma_t \varepsilon_t, \qquad \varepsilon_t \sim \mathcal{N}(0, 1),$$

$$\sigma_t^2 = \omega + \alpha x_{t-1}^2 + \beta \sigma_{t-1}^2.$$

Here $x_t$ represents heavy-tailed, volatility-clustered "returns", while $\sigma_t^2$ evolves as a GARCH(1,1) variance process.

### A.3.3. REAL-WORLD BENCHMARKS

**ETT (Electricity Transformer Temperature).** The ETT benchmark, introduced with Informer (Zhou et al., 2021), contains measurements from two electricity transformers over two years:

- ETTm1, ETTm2: 7 variables, 15-minute sampling, 69,680 time steps.

- ETTh1, ETTh2: 7 variables, 1-hour sampling, 17,420 time steps.

**Weather.** The Weather benchmark contains meteorological measurements recorded every 10 minutes in 2020 at the Max Planck Biogeochemistry Institute weather station in Jena, Germany; it is commonly used in LTSF benchmarks following Autoformer (Wu et al., 2021).

### A.4. App: Experiments Setup

**Training setup** For numerical convenience we use an isotropic base $z \sim \mathcal{N}(0, aI)$ and $y_0 \sim \mathcal{N}(0, aI)$ with a scalar $a > 0$; In both experiments we use $a = 0.1$; this simply rescales the eigenvalues in $\Lambda$ (equivalently $\tilde{A} = \sqrt{a}\, A$, $\Sigma = \tilde{A}\tilde{A}^\top$.) and does not change the SPD Jacobian structure or the optimal-transport interpretation.

We refer to the multi-horizon experiments as the *detailed table* and the main shock-evaluation experiments as the *shock table*. Detailed table and shock experiments use different settings (though *uniformly applied to all models* in respective experiments) to capture different aspects of model behavior. We discuss important points:

- **Optimizer:** all models are implemented in PyTorch and trained with AdamW (Loshchilov & Hutter, 2019) with no weight decay.

- **Learning rate:** the shock table uses $3 \times 10^{-4}$, and the detailed tables use $9 \times 10^{-4}$.

- **Epochs and patience:** both protocols use 50 epochs with patience = 5.

- **Batch size:** 95 for the shock table and 128 for the detailed tables.

- **Grace period:** validation is logged but cannot trigger early stopping for the first 3 epochs in the shock table; the detailed tables disable this grace period.

- **Training objective:** all models are trained only with Huber loss ($\delta = 1.0$).

- **Evaluation objective:** $0.1 \cdot \text{MSE} + 1.0 \cdot \text{MAE} + 0.1 \cdot \text{SWD/WD}$, with protocol-specific details below.

- **SWD vs. WD:** the shock table uses no random projections, reducing SWD to the 1D W2 metric reported as WD; the detailed tables use 500 random projections for SWD.

- **Validation smoothing:** no exponential smoothing of the validation objective is used.

- **Input length:** both protocols use input length 336.

- **Forecast horizon:** the shock table uses horizon 336; the detailed tables use horizons $\{96, 192, 336, 720\}$.

- **Seeds:** shock table $\{7, 1955\}$; detailed tables $\{7, 1955, 2023, 4\}$.

- **Data split:** shock table 70%/20%/10% (to make reconstruction and visualization on the test set easier); detailed tables 70%/10%/20%.

- **Preprocessing:** zero imputation/removal is enabled only for ETTh1 and ETTm1 in the shock table. No sentinel-value correction is used in either protocol.

- **Dataloader:** no dataset uses `drop_last`.

**Intro to Shock Experiments**   See Table 7 for detailed system parameters. We record the state after every integration step, so the sampling interval of the time series equals the solver step size $dt$. Since the train/validation/test split is 70/20/10, `shock_frac`$= 0.35$ means the shock occurs at 35% of the full trajectory, i.e. halfway through the training split.

| System | Scenario | $dt$ | steps | Parameters / shock description |
|---|---|---|---|---|
| *Main chaotic benchmarks (no shock).* | | | | |
| Lorenz-63 | main | 0.01 | 25000 | `sigma` = 10, `rho` = 28, `beta` = 8/3. |
| Rössler | main | 0.01 | 25000 | `a` = 0.2, `b` = 0.2, `c` = 5.7. |
| Chua | main | 0.005 | 35000 | `alpha` = 15.6, `beta` = 28.0, `m0` = −8/7, `m1` = −5/7. |
| *Synthetic shock scenarios.* | | | | |
| Lorenz-63 | LORENZ_BASE | 0.01 | 35999 | Baseline Lorenz-63, no shock; default `sigma` = 10, `rho` = 28, `beta` = 8/3. |
| Lorenz-63 | LORENZ_PARAM | 0.01 | 35999 | Parameter shock (`shock_kind` = "param"): `sigma` : 10 → 10.1, `rho` : 28 → 28.1, `beta` : 8/3 → 8.1/3. |
| Lorenz-63 | LORENZ_STATE | 0.01 | 35999 | State shock (`shock_kind` = "state_eps"): `shock_eps` = 0.9; ODE parameters as in LORENZ_BASE. |
| Lorenz-63 | LORENZ_SWITCH | 0.01 | 35999 | Switch shock (`shock_kind` = "switch"): `switch_update` sets `rho` : 28 → 28.1 and `initial_cond` : [1.0, 0.98, 1.1] → [1.002, 0.982, 1.102]. |
| Rössler | ROSSLER_BASE | 0.01 | 35999 | Baseline Rössler, no shock; `a` = 0.2, `b` = 0.2, `c` = 5.7. |
| Rössler | ROSSLER_PARAM | 0.01 | 35999 | Parameter shock (`shock_kind` = "param"): `a` : 0.2 → 0.25, `b` : 0.2 → 0.25, `c` : 5.7 → 5.75. |
| Lorenz-96 | LORENZ96_BASE | 0.007 | 55000 | Baseline Lorenz-96; `dim` = 6, `forcing` = 8.0, method=rk4. |
| Lorenz-96 | LORENZ96_SWITCH | 0.007 | 55000 | Switch shock (`shock_kind` = "switch"): `switch_update` sets `forcing` : 8.0 → 9.0 and `initial_cond` = [0.99, 1.02, 1.02, 1.03, 1.01, 1.01] (with `dim` = 6, method=rk4). |
| Chua | CHUA_BASE | 0.005 | 35999 | Baseline Chua, no shock; `alpha` = 15.6, `beta` = 28.0, `m0` = −8/7, `m1` = −5/7. |
| Chua | CHUA_PARAM | 0.005 | 35999 | Parameter shock (`shock_kind` = "param"): `alpha` : 15.6 → 15.9, `beta` : 28.0 → 28.5, `m0` : −8/7 → −8.1/7, `m1` : −5/7 → −5.2/7. |
| Chua | CHUA_SWITCH | 0.005 | 35999 | Switch shock (`shock_kind` = "switch"): `switch_update` sets `initial_cond` = [0.11, 0.01, 0.02]; other parameters as in CHUA_BASE. |
| OU | OU_BASE | 0.5 | 25000 | Baseline Ornstein–Uhlenbeck; `initial_cond` = [0.0], `theta` = 0.2, `mu` = 0.0, `sigma` = 0.3, method=method="euler". |
| OU | OU_PARAM | 0.5 | 25000 | Parameter shock (`shock_kind` = "param"): `mu` : 0.0 → 0.5; other OU parameters as in OU_BASE. |
| SLDS | SLDS_BASE | 0.01 | 25000 | Baseline switching linear dynamical system; A1 = 0.9, Q1 = 0.05, A2 = 0.98, Q2 = 0.35, p11 = 0.94, p22 = 0.95. |
| SLDS | SLDS_PARAM | 0.01 | 25000 | Parameter shock (`shock_kind` = "param"): A1 : 0.9 → 0.83, Q1 : 0.05 → 0.50, A2 : 0.98 → 0.97, Q2 : 0.35 → 0.30, p11 : 0.94 → 0.96, p22 : 0.95 → 0.92. |
| SLDS | SLDS_SWITCH | 0.01 | 25000 | Switch shock (`shock_kind` = "switch"): `switch_update` sets A1 = 0.87, Q1 = 0.07, A2 = 0.99, Q2 = 0.45, p11 = 0.90, p22 = 0.95. |
| Double-well | DOUBLEWELL_BASE | 0.5 | 25000 | Baseline double-well SDE (Euler API, step size 0.5); `a` = 1.5, `sigma` = 0.25, seed = 1955. |
| Double-well | DOUBLEWELL_PARAM | 0.5 | 25000 | Parameter shock (`shock_kind` = "param"): `a` : 1.5 → 1.0, `sigma` : 0.25 → 0.35. |
| Double-well | DOUBLEWELL_SWITCH | 0.5 | 25000 | Switch shock (`shock_kind` = "switch"): `switch_update` sets `a` = 1.0, `sigma` = 0.35 (same target as DOUBLEWELL_PARAM). |
| Seasonal AR | SEASONAL_AR_BASE | 0.01 | 25000 | Baseline seasonal AR process (discrete-time; `dt`/method kept for API): S = 24, `phi` = 0.5, `sigma` = 0.2, a0 = 1.0, `amp_drift_per_step` = 0. |
| Seasonal AR | SEASONAL_AR_PARAM | 0.01 | 25000 | Parameter shock (`shock_kind` = "param"): a0 : 1.0 → 1.4, `sigma` : 0.2 → 0.35, `phi` : 0.5 → 0.8. |
| GARCH(1,1) | GARCH_BASE | 0.01 | 25000 | Baseline GARCH(1,1) volatility model (discrete-time; `dt`/method kept for API): omega = 0.01, alpha = 0.06, beta = 0.90. |
| GARCH(1,1) | GARCH_PARAM | 0.01 | 25000 | Parameter shock (`shock_kind` = "param"): omega : 0.01 → 0.03, alpha : 0.06 → 0.15, beta : 0.90 → 0.70. |
| KS | KS_BASE | 0.01 | 25000 | Baseline Kuramoto–Sivashinsky; nx = 64, Lx = 22.0, nu = 1.0, method="etdrk4". |
| KS | KS_PARAM | 0.01 | 25000 | Parameter shock (`shock_kind` = "param"): nu : 1.0 → 0.80 (other KS parameters as in KS_BASE). |

*Table 7.* Parameter settings for the main chaotic benchmarks (top block) and all available generators (bottom block). Shock scenarios are instantiated via `PremadeID.*` in `make_source`, with `shock_frac` fixed to 0.35 so that shocks are applied after the first 35% of the trajectory.

## A.5. App: Ablation, Footprint and Detail Tables

**Protocol note.** The detailed tables, shock table, ablation table, and compute-footprint table are reported under their own protocols. Each table is internally consistent within its regime, but the detailed and shock tables use different settings; we therefore avoid direct quantitative comparisons across regimes unless models are rerun under a matched setup. The released repository linked in the main text contains the shock-table code snapshot and configuration records, together with archived metrics/logs and protocol notes for the detailed tables, ablations, and compute-footprint results.

| Variant (dataset) | MSE↓ | MAE↓ | WD↓ | EPT↑ | Time (s) |
|---|---|---|---|---|---|
| Base (ETTh1) | 10.96 | 1.88 | 5.75 | 63.34 | — |
| Base (Lorenz-63) | **21.66** | **2.39** | **4.41** | 241.25 | — |
| No encoder & no mean updates (Lorenz-63) | 194.43 | 10.99 | 189.47 | 17.10 | 492.73 |
| Only encoder (ETTh1) | 11.17 | 1.87 | 5.86 | 63.00 | 378.93 |
| Only encoder (Lorenz-63) | 27.09 | 2.85 | 6.17 | 214.10 | 1127.98 |
| No rotation (ETTh1) | 11.84 | 1.90 | 7.72 | 66.00 | 617.55 |
| No rotation (Lorenz-63) | 27.62 | 2.94 | 6.19 | 203.60 | 846.38 |
| No patching (ETTh1) | 10.99 | **1.84** | 5.38 | 64.10 | 422.65 |
| No patching (Lorenz-63) | 22.86 | 2.50 | 4.47 | 231.10 | 1239.11 |
| Reflection = 2 (ETTh1) | 11.52 | 1.89 | 6.02 | 64.30 | 429.67 |
| Reflection = 2 (Lorenz-63) | 26.14 | 2.79 | 6.51 | 199.10 | 889.23 |
| Reflection = 24 (8-block) (ETTh1) | **10.91** | 1.87 | **5.21** | 63.00 | 432.86 |
| Reflection = 24 (8-block) (Lorenz-63) | 23.92 | 2.70 | 5.70 | 213.00 | 930.42 |

*Table 8.* Ablations of Fern components under the shock-table-style 336-input/336-forecast protocol on ETTh1 and Lorenz-63. Base uses $R = 8$ reflections and patch size $p = 24$; **bold** marks the best MSE, MAE, and WD. "No encoder & no mean updates" disables both the bidirectional encoder and output translation updates. "8-block" denotes block-update training for the 24-reflection setting.

| Data | fr | | | tm | | | tst | | | dl | | |
|---|---|---|---|---|---|---|---|---|---|---|---|---|
| | MSE | MAE | WD | MSE | MAE | WD | MSE | MAE | WD | MSE | MAE | WD |
| Lorenz | **21.82**$_{\pm 2.13}$ | **2.17** | **2.23** | 30.94$_{\pm 5.62}$ | 3.19 | 11.11 | 30.11$_{\pm 2.92}$ | 3.28 | 9.60 | 67.76$_{\pm 1.12}$ | 6.07 | 38.22 |
| Rossler | **0.04**$_{\pm 0.01}$ | **0.11** | **0.02** | 6.01$_{\pm 0.26}$ | 1.09 | 5.20 | 8.33$_{\pm 0.36}$ | 1.43 | 7.25 | 11.64$_{\pm 0.45}$ | 1.82 | 10.20 |
| Chua | **0.08**$_{\pm 0.13}$ | **0.08** | **0.05** | 0.20$_{\pm 0.21}$ | 0.16 | 0.15 | 0.49$_{\pm 0.13}$ | 0.32 | 0.37 | 0.39$_{\pm 0.02}$ | 0.30 | 0.24 |
| ETTh1 | **6.60**$_{\pm 0.11}$ | 1.53 | **2.64** | 6.83$_{\pm 0.16}$ | **1.52** | 2.83 | 6.62$_{\pm 0.14}$ | 1.54 | 2.77 | 7.04$_{\pm 0.06}$ | 1.53 | 2.75 |
| ETTm1 | 5.80$_{\pm 0.25}$ | 1.45 | 2.85 | **5.27**$_{\pm 0.22}$ | 1.39 | 2.60 | 5.36$_{\pm 0.42}$ | **1.37** | **2.44** | 6.31$_{\pm 0.11}$ | 1.39 | 3.65 |

*Table 9.* Aggregated errors across prediction horizons $H \in \{96, 192, 336, 720\}$ with input length=336. Models: **fr** = Fern, **tm** = TimeMixer, **tst** = PatchTST, **dl** = DLinear. Values are means over 4 seeds [7, 1955, 2023, 4]; tiny ± indicates standard error computed from per-horizon standard errors (see text). Best and second-best values for each row/metric are highlighted.

| Model | Dataset | Training time | Params (M) | GFLOPs (G) |
|---|---|---|---|---|
| Fern | L63 | 960 | 1.025 | 0.0035 |
| TimeMixer | L63 | 1280 | 0.886 | 0.0463 |
| PatchTST | L63 | 600 | 2.008 | 0.0308 |
| DLinear | L63 | 150 | 0.679 | 0.0007 |
| Fern | m1 | 83 | 1.025 | 0.0035 |
| TimeMixer | m1 | 120 | 0.886 | 0.0463 |
| PatchTST | m1 | 110 | 2.008 | 0.0308 |
| DLinear | m1 | 27 | 0.679 | 0.0007 |

*Table 10.* Compute footprint. Training time is reported in seconds for Lorenz-63 (336-in-336-out, 52k steps) and ETTm1 (96-in-336-out). GFLOPs are reported per sample per step.

| Data | Hor. | Model | MSE | MAE | SWD | EPT |
|------|------|-------|-----|-----|-----|-----|
| Chua | 96 | Fern | $0.0007 \pm 0.00$ | $0.0191 \pm 0.01$ | $0.0007 \pm 0.00$ | 96 |
| | | TimeMixer | $0.0015 \pm 0.00$ | $0.0259 \pm 0.00$ | $0.0014 \pm 0.00$ | 96 |
| | | PatchTST | $0.0063 \pm 0.00$ | $0.0509 \pm 0.01$ | $0.0057 \pm 0.00$ | 96 |
| | | DLinear | $0.0168 \pm 0.00$ | $0.0796 \pm 0.01$ | $0.0167 \pm 0.00$ | 96 |
| | 192 | Fern | $0.0011 \pm 0.00$ | $0.0257 \pm 0.00$ | $0.0009 \pm 0.00$ | 192 |
| | | TimeMixer | $0.0349 \pm 0.01$ | $0.1199 \pm 0.01$ | $0.0319 \pm 0.01$ | 192 |
| | | PatchTST | $0.1242 \pm 0.06$ | $0.1924 \pm 0.05$ | $0.1091 \pm 0.05$ | 192 |
| | | DLinear | $0.0265 \pm 0.01$ | $0.1183 \pm 0.02$ | $0.0246 \pm 0.01$ | 192 |
| | 336 | Fern | $0.0010 \pm 0.00$ | $0.0225 \pm 0.00$ | $0.0004 \pm 0.00$ | 336 |
| | | TimeMixer | $0.0045 \pm 0.00$ | $0.0479 \pm 0.01$ | $0.0028 \pm 0.00$ | 336 |
| | | PatchTST | $0.0278 \pm 0.01$ | $0.1060 \pm 0.02$ | $0.0215 \pm 0.01$ | 336 |
| | | DLinear | $0.1590 \pm 0.01$ | $0.2703 \pm 0.01$ | $0.0813 \pm 0.01$ | 336 |
| | 720 | Fern | $0.3301 \pm 0.51$ | $0.2386 \pm 0.17$ | $0.1994 \pm 0.33$ | $474 \pm 73$ |
| | | TimeMixer | $0.7624 \pm 0.82$ | $0.4475 \pm 0.26$ | $0.5685 \pm 0.64$ | $427 \pm 80$ |
| | | PatchTST | $1.7924 \pm 0.52$ | $0.9209 \pm 0.11$ | $1.3499 \pm 0.43$ | $228 \pm 45$ |
| | | DLinear | $1.3719 \pm 0.09$ | $0.7314 \pm 0.03$ | $0.8305 \pm 0.07$ | $119 \pm 2$ |
| | *Avg* | Fern | 0.0832 | 0.0765 | 0.0504 | 274.4 |
| | | TimeMixer | 0.20083 | 0.16030 | 0.15115 | 262.7 |
| | | PatchTST | 0.48767 | 0.31755 | 0.37155 | 213.0 |
| | | DLinear | 0.39355 | 0.29990 | 0.23828 | 185.6 |

*Table 11.* Chua's circuit ($dt = 5 \times 10^{-3}$, 35,000 steps). Values are mean $\pm$ s.e. across 4 seeds. Higher is better for EPT; lower is better for all other metrics.

| Data | Hor. | Model | MSE | MAE | SWD | EPT |
|------|------|-------|-----|-----|-----|-----|
| Rossler | 96 | Fern | $0.0032 \pm 0.00$ | $0.0445 \pm 0.01$ | $0.0030 \pm 0.00$ | 96 |
| | | TimeMixer | $0.1033 \pm 0.07$ | $0.2454 \pm 0.09$ | $0.0963 \pm 0.07$ | 96 |
| | | PatchTST | $0.1846 \pm 0.10$ | $0.3155 \pm 0.09$ | $0.1700 \pm 0.10$ | 96 |
| | | DLinear | $0.3036 \pm 0.09$ | $0.3343 \pm 0.06$ | $0.2908 \pm 0.08$ | 96 |
| | 192 | Fern | $0.0066 \pm 0.00$ | $0.0618 \pm 0.01$ | $0.0055 \pm 0.00$ | 192 |
| | | TimeMixer | $0.1124 \pm 0.03$ | $0.2689 \pm 0.03$ | $0.1016 \pm 0.03$ | 192 |
| | | PatchTST | $0.5704 \pm 0.14$ | $0.5893 \pm 0.09$ | $0.4762 \pm 0.14$ | 192 |
| | | DLinear | $3.6638 \pm 1.14$ | $1.0719 \pm 0.19$ | $3.2148 \pm 1.11$ | $179 \pm 8$ |
| | 336 | Fern | $0.0830 \pm 0.03$ | $0.1988 \pm 0.04$ | $0.0567 \pm 0.02$ | $332 \pm 4$ |
| | | TimeMixer | $14.9881 \pm 0.84$ | $2.4916 \pm 0.09$ | $12.8596 \pm 0.74$ | $162 \pm 3$ |
| | | PatchTST | $23.9804 \pm 1.43$ | $3.3559 \pm 0.12$ | $20.8028 \pm 1.51$ | $69 \pm 1$ |
| | | DLinear | $34.3263 \pm 1.34$ | $4.3221 \pm 0.09$ | $29.8861 \pm 1.22$ | $60 \pm 1$ |
| | 720 | Fern | $0.0548 \pm 0.04$ | $0.1258 \pm 0.03$ | $0.0160 \pm 0.01$ | $677 \pm 44$ |
| | | TimeMixer | $8.8210 \pm 0.60$ | $1.3566 \pm 0.06$ | $7.7532 \pm 0.47$ | $557 \pm 21$ |
| | | PatchTST | $8.5990 \pm 0.19$ | $1.4545 \pm 0.02$ | $7.5545 \pm 0.16$ | $539 \pm 3$ |
| | | DLinear | $8.2653 \pm 0.34$ | $1.5395 \pm 0.09$ | $7.3939 \pm 0.29$ | $594 \pm 24$ |
| | *Avg* | Fern | 0.0369 | 0.1077 | 0.0203 | 324.2 |
| | | TimeMixer | 6.0062 | 1.0906 | 5.2027 | 251.8 |
| | | PatchTST | 8.3336 | 1.4288 | 7.2509 | 223.9 |
| | | DLinear | 11.6398 | 1.8170 | 10.1964 | 232.4 |

*Table 12.* Rössler ($dt = 10^{-2}$, 25,000 steps). Values are mean $\pm$ s.e. across 4 seeds. Higher is better for EPT; lower is better for all other metrics.

| Data | Hor. | Model | MSE | MAE | SWD | EPT |
|------|------|-------|-----|-----|-----|-----|
| ETTh1 | 96 | Fern | $6.68 \pm 0.05$ | $1.50 \pm 0.01$ | $2.88 \pm 0.09$ | $34.11 \pm 0.68$ |
| | | TimeMixer | $6.85 \pm 0.43$ | $1.47 \pm 0.04$ | $3.02 \pm 0.28$ | $39.40 \pm 1.31$ |
| | | PatchTST | $5.87 \pm 0.14$ | $1.43 \pm 0.03$ | $2.48 \pm 0.08$ | $41.45 \pm 1.15$ |
| | | DLinear | $6.62 \pm 0.06$ | $1.45 \pm 0.01$ | $2.77 \pm 0.10$ | $38.53 \pm 1.15$ |
| | 192 | Fern | $6.21 \pm 0.19$ | $1.49 \pm 0.03$ | $1.81 \pm 0.13$ | $61.29 \pm 7.42$ |
| | | TimeMixer | $6.82 \pm 0.08$ | $1.49 \pm 0.02$ | $2.21 \pm 0.10$ | $70.84 \pm 4.58$ |
| | | PatchTST | $6.12 \pm 0.24$ | $1.48 \pm 0.03$ | $1.81 \pm 0.11$ | $78.49 \pm 8.21$ |
| | | DLinear | $7.39 \pm 0.17$ | $1.51 \pm 0.03$ | $2.36 \pm 0.17$ | $81.34 \pm 6.13$ |
| | 336 | Fern | $6.41 \pm 0.33$ | $1.54 \pm 0.03$ | $3.25 \pm 0.34$ | $68.77 \pm 7.36$ |
| | | TimeMixer | $6.06 \pm 0.20$ | $1.43 \pm 0.01$ | $3.27 \pm 0.28$ | $72.88 \pm 1.20$ |
| | | PatchTST | $6.81 \pm 0.47$ | $1.56 \pm 0.05$ | $3.75 \pm 0.63$ | $69.49 \pm 1.49$ |
| | | DLinear | $6.39 \pm 0.05$ | $1.48 \pm 0.01$ | $3.16 \pm 0.11$ | $74.75 \pm 2.14$ |
| | 720 | Fern | $7.09 \pm 0.20$ | $1.57 \pm 0.06$ | $2.60 \pm 0.19$ | $112.16 \pm 18.48$ |
| | | TimeMixer | $7.57 \pm 0.45$ | $1.69 \pm 0.07$ | $2.80 \pm 0.35$ | $125.88 \pm 2.06$ |
| | | PatchTST | $7.67 \pm 0.14$ | $1.70 \pm 0.05$ | $3.03 \pm 0.15$ | $121.38 \pm 1.47$ |
| | | DLinear | $7.77 \pm 0.15$ | $1.67 \pm 0.05$ | $2.70 \pm 0.11$ | $124.28 \pm 0.78$ |
| *Simple Average* | | Fern | 6.60 | 1.53 | 2.64 | 69.08 |
| | | TimeMixer | 6.83 | 1.52 | 2.83 | 77.25 |
| | | PatchTST | 6.62 | 1.54 | 2.77 | 77.70 |
| | | DLinear | 7.04 | 1.53 | 2.75 | 79.72 |

*Table 13.* ETTh1. Values are mean $\pm$ s.e. across 4 seeds. Higher is better for EPT, lower is better for the rest.

| Data | Hor. | Model | MSE | MAE | SWD | EPT |
|------|------|-------|-----|-----|-----|-----|
| ETTm1 | 96 | Fern | $2.67 \pm 0.03$ | $1.07 \pm 0.03$ | $0.95 \pm 0.08$ | $44.32 \pm 1.27$ |
| | | TimeMixer | $2.91 \pm 0.64$ | $0.98 \pm 0.09$ | $1.07 \pm 0.45$ | $46.79 \pm 2.80$ |
| | | PatchTST | $3.03 \pm 0.36$ | $1.04 \pm 0.06$ | $1.14 \pm 0.31$ | $43.57 \pm 0.35$ |
| | | DLinear | $2.33 \pm 0.11$ | $0.91 \pm 0.03$ | $1.01 \pm 0.11$ | $43.32 \pm 2.22$ |
| | 192 | Fern | $6.86 \pm 0.69$ | $1.59 \pm 0.06$ | $5.30 \pm 0.58$ | $66.75 \pm 1.64$ |
| | | TimeMixer | $5.77 \pm 0.48$ | $1.62 \pm 0.09$ | $4.32 \pm 0.38$ | $69.97 \pm 4.67$ |
| | | PatchTST | $6.60 \pm 1.60$ | $1.54 \pm 0.13$ | $4.96 \pm 1.45$ | $67.17 \pm 2.99$ |
| | | DLinear | $7.10 \pm 0.33$ | $1.50 \pm 0.04$ | $5.60 \pm 0.36$ | $69.99 \pm 1.90$ |
| | 336 | Fern | $6.42 \pm 0.56$ | $1.57 \pm 0.05$ | $3.45 \pm 0.72$ | $105.69 \pm 4.66$ |
| | | TimeMixer | $6.04 \pm 0.32$ | $1.55 \pm 0.08$ | $3.61 \pm 0.41$ | $107.72 \pm 3.85$ |
| | | PatchTST | $5.75 \pm 0.22$ | $1.45 \pm 0.06$ | $2.31 \pm 0.50$ | $101.94 \pm 4.42$ |
| | | DLinear | $7.98 \pm 0.23$ | $1.56 \pm 0.04$ | $5.37 \pm 0.28$ | $111.09 \pm 0.47$ |
| | 720 | Fern | $7.26 \pm 0.43$ | $1.58 \pm 0.05$ | $1.68 \pm 0.18$ | $162.74 \pm 16.46$ |
| | | TimeMixer | $6.35 \pm 0.15$ | $1.41 \pm 0.03$ | $1.39 \pm 0.28$ | $207.69 \pm 15.89$ |
| | | PatchTST | $6.06 \pm 0.14$ | $1.44 \pm 0.02$ | $1.35 \pm 0.22$ | $212.47 \pm 9.95$ |
| | | DLinear | $7.82 \pm 0.07$ | $1.58 \pm 0.00$ | $2.61 \pm 0.15$ | $186.79 \pm 13.50$ |
| | *Avg* | Fern | 5.80 | 1.45 | 2.85 | 94.88 |
| | | TimeMixer | 5.27 | 1.39 | 2.60 | 108.04 |
| | | PatchTST | 5.36 | 1.37 | 2.44 | 106.29 |
| | | DLinear | 6.31 | 1.39 | 3.65 | 102.80 |

*Table 14.* ETTm1. Values are mean $\pm$ s.e. across 4 seeds. Higher is better for EPT, lower is better for the rest.

| Data | Hor. | Model | MSE | MAE | SWD | EPT |
|------|------|-------|-----|-----|-----|-----|
| Lorenz | 96 | Fern | $0.47 \pm 0.15$ | $0.50 \pm 0.08$ | $0.21 \pm 0.06$ | $96.00 \pm 0.00$ |
| | | TimeMixer | $0.18 \pm 0.01$ | $0.33 \pm 0.01$ | $0.10 \pm 0.01$ | $96.00 \pm 0.00$ |
| | | PatchTST | $1.17 \pm 0.31$ | $0.83 \pm 0.09$ | $0.86 \pm 0.22$ | $96.00 \pm 0.00$ |
| | | DLinear | $54.04 \pm 3.03$ | $4.92 \pm 0.17$ | $33.48 \pm 1.79$ | $50.42 \pm 2.88$ |
| | 192 | Fern | $2.06 \pm 0.69$ | $0.83 \pm 0.13$ | $0.33 \pm 0.09$ | $175.23 \pm 6.90$ |
| | | TimeMixer | $16.91 \pm 16.46$ | $2.45 \pm 1.20$ | $8.55 \pm 8.93$ | $136.16 \pm 56.75$ |
| | | PatchTST | $6.90 \pm 2.06$ | $1.91 \pm 0.29$ | $2.52 \pm 0.80$ | $179.67 \pm 12.43$ |
| | | DLinear | $79.25 \pm 2.61$ | $6.78 \pm 0.15$ | $48.51 \pm 1.24$ | $19.76 \pm 0.18$ |
| | 336 | Fern | $21.25 \pm 8.18$ | $2.39 \pm 0.41$ | $3.49 \pm 1.38$ | $187.03 \pm 12.57$ |
| | | TimeMixer | $55.97 \pm 2.09$ | $5.10 \pm 0.07$ | $25.41 \pm 1.99$ | $92.78 \pm 7.91$ |
| | | PatchTST | $60.96 \pm 9.40$ | $5.43 \pm 0.44$ | $25.12 \pm 2.81$ | $105.74 \pm 14.05$ |
| | | DLinear | $61.06 \pm 1.31$ | $5.64 \pm 0.12$ | $30.28 \pm 0.37$ | $79.80 \pm 1.02$ |
| | 720 | Fern | $63.52 \pm 2.25$ | $4.96 \pm 0.20$ | $4.89 \pm 0.57$ | $293.46 \pm 14.90$ |
| | | TimeMixer | $50.71 \pm 15.19$ | $4.91 \pm 0.40$ | $10.39 \pm 3.56$ | $247.14 \pm 13.47$ |
| | | PatchTST | $51.41 \pm 6.62$ | $4.94 \pm 0.24$ | $9.89 \pm 2.96$ | $254.76 \pm 29.58$ |
| | | DLinear | $76.69 \pm 1.48$ | $6.96 \pm 0.11$ | $40.59 \pm 0.29$ | $24.01 \pm 3.50$ |
| | *Avg* | Fern | 21.82 | 2.17 | 2.23 | 187.93 |
| | | TimeMixer | 30.94 | 3.19 | 11.11 | 143.02 |
| | | PatchTST | 30.11 | 3.28 | 9.60 | 159.04 |
| | | DLinear | 67.76 | 6.07 | 38.22 | 43.50 |

*Table 15.* Lorenz-63. Values are mean $\pm$ s.e. across 4 seeds. Higher is better for EPT; lower is better for all other metrics.

