# OpenReview forum: "Ellipsoidal Time Series Forecasting"
_ICML.cc/2026/Conference — ICML 2026 regular_

### Official Review · Reviewer_iNGy · 2026-03-07

**Soundness:** 3
**Presentation:** 3
**Significance:** 3
**Originality:** 3
**Overall Recommendation:** 4
**Confidence:** 2

**Summary:**

The paper analyzes long-term time series forecasting and models system dynamics through local geometric structure rather than only conditional mean prediction. Overall, this study's specific aspect pertains to learning local Jacobians with explicit spectral structure by framing forecasting as an optimal transport problem.

The proposed model, Fern, transports samples from a Gaussian source to data-dependent Gaussian ellipsoids using a symmetric positive semi-definite Jacobian parameterized through its spectral components. Experiments show improved robustness to chaotic dynamics and non-stationary shocks compared to several forecasting baselines.

**Compliance With Llm Reviewing Policy:**

Affirmed.

**Final Justification:**

My questions have been addressed during the rebuttal. I confirm my positive score.

**Key Questions For Authors:**

- How sensitive is performance to the choice of patch size and number of reflections?

- How does the approach scale when the number of variables is large? (experiments mainly use moderate-dimensional systems)

**Limitations:**

yes

**Strengths And Weaknesses:**

Strengths:
- Interesting geometric formulation: novel view of time-series forecasting as conditional manifold transport. The use of optimal transport and Brenier’s theorem to constrain Jacobians to the SPD class is conceptually elegant and provides a solid theoretical motivation.
- Computational efficiency through spectral parameterization: the proposed parameterization replaces cubic-time eigendecomposition with a linear-time spectral search.
- Authors highlight weaknesses in existing LTSF benchmarks and introduce controlled synthetic experiments based on dynamical systems. This is a valuable direction to advance current forecasting benchmarks to test robustness to regime shifts or chaotic dynamics.
- Extensive experiments on synthetic systems: the experimental section evaluates the model across a large set of dynamical systems (chaotic, stochastic, switching, etc.), which provides a convincing demonstration of robustness in those settings.

Weaknesses:
- Although Brenier’s theorem motivates the SPD parameterization, the paper does not provide theoretical results about generalization
or approximation capacity of the proposed architecture.
- Although the spectral parameterization is efficient relative to naive Jacobian modeling, the full architecture still includes several nontrivial components (encoder, reflections, patching). It would be helpful to better quantify runtime and memory costs compared to simpler baselines (even empirically).

---

> ### Author Rebuttal · Authors · 2026-03-27
>
> This year's ICML reviewing process is long and tiring, involving a substantial amount of work. We take this chance to sincerely thank the reviewer for the hard-work and service to the community. We address the weaknesses:
>
> **"Although Brenier’s theorem motivates the SPD parameterization, the paper does not provide theoretical results about generalization or approximation capacity of the proposed architecture."**
>
> We acknowledge that generalization bounds for OT-parameterized architectures are an open challenge in statistical learning theory, placing it out of scope for this specific paper. However, on the approximation capacity side, we do have some concrete results that that we touched on in the paper:
>
> -Householder products can (in fact, both sufficient and necessary) represent *any* orthogonal matrix in the patch with R(# of Householder reflection) = P(patch size); when $R < P$, this represents a reduced capacity search.
> -SPD factorization can represent any SPD covariance/Jacobian in the patch, therefore, Fern can represent *any* Gaussian-to-Gaussian affine OT map patchwise
> -Empirically, we have investigated how different $R$ affects performances; Table 8 shows that as $R$ increases to $P$ (patch size), additional rotational capacity is no longer the bottleneck and performance saturates. In addition, Table 9 shows stable performance across 4 seeds with tight standard errors.
>
> **"Although the spectral parameterization is efficient relative to naive Jacobian modeling, the full architecture still includes several nontrivial components (encoder, reflections, patching). It would be helpful to better quantify runtime and memory costs compared to simpler baselines (even empirically)."**
>
> We provided a "Compute footprint" table (Table 10) comparing Fern to TimeMixer, PatchTST, and DLinear across Training Time, Params, and GFLOPs. For example, on ETTm1, Fern takes 83 seconds to train vs. TimeMixer's 120 seconds and PatchTST's 110 seconds.
>
> From an optimization point of view, the encoder is less of a problem, as its bidirectional affine coupling layer (ACL) structure is well-known and widely used in the Normalizing Flow literature. Householder reflection's main bottleneck is memory; the reflection itself is just a dot product and a fused multiply-add per step. To ensure we can use a higher number of reflections without encountering memory problems, we designed two methods to make it easy to use the *maximum number of R*:
>
> (1) block update: for example, we can use 24 reflections but only update 8 of them during a single pass, saving memory. On Lorenz63, 25k samples, we don't see a large difference in training time:
>
> Reflection: 8
> MSE=1.109 | MAE=0.570
> time: 484.86 seconds
>
> &nbsp;
>
> Reflection: 24 train with 3 size-8 blocks (updates one third each forward)
> MSE=1.286 | MAE=0.628 (expected: each block receives 1/3 the gradient updates per epoch; with longer training the gap closes)
> time: 457.08 seconds
>
> (2) (implemented in code) we write Triton code to use this trick: since every Householder reflection is involutory ($H^2 = I$, i.e. $H = H^{-1}$), applying the same reflection to $x_r$ recovers $x_{r-1}$. We can save the final output and apply reflection in reverse: instead of saving R intermediate vectors per patch ($O(R*P)$ memory), we do $R$ extra reflection applications during backward (just a dot product and a fused multiply-add per step). Memory cost stays $O(P)$ per patch regardless of R.
>
> These two methods allow us to reduce memory usage and increase the number of reflections, at the expense of slower update *or* extra reflections.
>
> **"How sensitive is performance to the choice of patch size and number of reflections?"**
>
> Our experience is that performance is stable across a wide range of $R$ and $R=8$ is a sweet spot for the tested datasets. Using the two approaches above, we can confidently set $R=P$ to avoid extra tuning. The only failure mode we noticed is when P is set too small that the intrinsic dim is larger than P; in the paper, systems like Lorenz have 3 dims, and Takens theorem works very well with delay-embedding with >10 dims. We use a conservative 24 dim as default; empirically, setting 48 dim yields little performance differences.
>
> **How does the approach scale when the number of variables is large? (experiments mainly use moderate-dimensional systems)**
>
> Fern is channel-independent by design, so computational cost scales strictly linearly ($O(c)$ with the number of variables $c$. The per-channel cost is $O(Rn)$ which depends only on horizon length $n$ and number of Householder reflections $R$. So scaling to hundreds of channels adds no architectural complexity. And this is a deliberate choice grounded in Takens' theorem (Section 3): each channel's delay embedding already reconstructs the attractor, so aggressive cross-channel mixing can be unnecessary and potentially harmful and we opt for the conservative, scalable CI design.

---

> > ### Author Rebuttal · Reviewer_iNGy · 2026-04-01
> >
> > Thanks for your answer. I keep my score.

---

### Official Review · Reviewer_91rW · 2026-03-11

**Soundness:** 1
**Presentation:** 1
**Significance:** 2
**Originality:** 1
**Overall Recommendation:** 2
**Confidence:** 5

**Summary:**

The article recasts the forecasting problem as an optimal transport one, and then exploits Brenier's theorem to reparametrise the transport using the fact that it corresponds to the gradient of a convex function. Experimental validations consider several datasets (only a single real-world dataset) and both the MSE and Wasserstein distance performance metrics.

**Compliance With Llm Reviewing Policy:**

Affirmed.

**Final Justification:**

Some of my concerns and misreadings of the results have been resolved, yet I still think this paper is not at a publication standard.

**Key Questions For Authors:**

Please see my comments above

**Strengths And Weaknesses:**

Despite the strong predictive performance claimed in Table 2, the paper has serious flaws, which mainly relate to the presentation and delivery of ideas.

 - The paper is written in a rather informal way, with frequent imprecise statements. For instance:
   - "we believe that conditional manifold transport is the core task of time series forecasting",
   - "move away from general $x$ to $y$ time-evolution paradigm",
   - referring to the contribution as "our trick",
   - explaining Brenier theorem as an "obvious fact",
   - "this inspires a worldview seeing data-generating process",
   - "$\Lambda$ is an $n$-dim vector with $O(n)$ cost"
- The problem is formulated, initially, for a 100-dimensional time series, with the first 70 values considered as observations. Why not a general length? The first page refers to a computational cost that is cubic in $n$, but $n$ was never defined
- The numbered steps on page 3 (left column) are hard to follow, mainly due to the missing punctuation
- There is a critical need for missing references: In the first page, there is extensive use of non-standard concepts such as time-delay embeddings, Lyapunov time, stochastic noise, and deterministic chaos. There are several references to the concept of a manifold that is not supported by references either.
- In terms of formatting, the arbitrary use of bold and italic fonts makes it difficult to capture the relevant concepts.
- Probably the most critical part is Section 3: it spans more than 4 pages (more than 50% of the paper) without subsections. One can hardly identify parts related to evaluation criteria, the dataset, and even the empirical results.
- The presentation of the results is also below the expected bar:
   - Table 1 alternates between validation and test results, making it rather difficult to assess what the superior method is.
   - Fig 3 shows a ground truth and a prediction, but it is not said what method is being shown.
   - There are no provided error bars in the experiments.

Lastly, conceptually, the article fails to explain the method/architecture in detail, despite the extensive narrative. In most of the paper, the focus is on the Wasserstein-based component (gradient parametrisation), but this concept is rather general and can be applied to any architecture. It is only on page 3, succinctly before Algorithm 1, where there's a reference to the model architecture using an encoder $H$. This is then explored in the ablation study in Table 3; however, these results are inconclusive: for one dataset, the base model works best, but for the other dataset, it is the full model that works best.

---

> ### Author Rebuttal · Authors · 2026-03-26
>
> Response to Reviewer 91rW
>
> We understand ICML reviewing is a long and tiring process, and we appreciate you taking the time to provide feedback. However, we must respectfully correct several factual misreadings regarding the manuscript's empirical reporting, mathematical definitions, and architectural descriptions.
>
> 1. Factual Corrections on Empirical Reporting
>
> "There are no provided error bars": This is incorrect. Tables 9, 11, 12, 13, 14, and 15 report means and standard errors across 4 random seeds for all tested horizons.
>
> "Table 1 alternates between validation and test... making it difficult to assess": This pairing *is* the entire central argument of that section. The table explicitly demonstrates that validation and test errors severely *diverge* under mainstream protocols; early validation error is a trap, making it anti-predictive of final test performance.
>
>
> "Fig 3... it is not said what method is being shown": The text immediately referencing the figure explicitly names the method: "...prior to step 1000, Fern predicts values that appear wrong...".
>
> "Ablation results are inconclusive": Table 8 contains the full ablation metrics. It confirms that the 24-reflection configuration monotonically improves MSE, SWD, and EPT across all datasets compared to zero-rotation or zero-patching baselines.
>
> 2. Clarifications on Architecture and Mathematical Formulation
>
>
> "The problem is formulated initially for a 100-dimensional time series... why not a general length? ... n was never defined": The 100-step split is explicitly introduced as an **illustrative example** to build intuition. The general variable $n$ is defined immediately afterward: "requires n^2 Jacobian entries for an n-dimensional horizon". Our detailed experiments test horizons of 96, 192, 336, and 720.
>
> "Fails to explain the method/architecture in detail": Section 2 details the *exact* parameterization. Algorithm 1 provides the complete, step-by-step forward pass, including the bidirectional encoder and the optimal transport head.
>
> 3. Clarifications on Terminology and Citations
>
>
> "Missing references for time-delay embeddings... non-standard concepts such as stochastic noise and deterministic chaos": Chaos and stochasticity are foundational, widely known concepts in dynamical systems. Time-delay embeddings itself is a standard terminology, its related literature are explicitly cited in Section 3 using the foundational texts: Takens (1981), Stark et al. (1999), and Stark (2003).
>
> "Explaining Brenier theorem as an obvious fact": We did *not* call the theorem itself obvious. We said: "Intuitively, this is stating an obvious fact: the dynamics must push the 30-dim x into some distribution in the 70-dim Y space; for that particular distribution, **there must be a cost-minimal way to move a generic 70-dim N(0, I) distribution into that distribution**." We stated that the physical intuition behind it, that is, that there must be a cost-minimal way to transport mass from a noise distribution to a target distribution. We call *this* intuitive, "obvious fact" to help readers grasp the math. The theorem itself was defined at the beginning of Section 2.
>
> "Referring to the contribution as 'our trick'": This is standard machine learning nomenclature, akin to the universally used "reparameterization trick" in variational inference. Here is a DeepMind paper with this word in its title, see: https://arxiv.org/pdf/2105.05347
>
> As for the listed sentences: when we say "we believe that conditional manifold transport is the core task of time series forecasting...": the precise statement in the paper is "When the manifold is as simple as a stable sine wave extending through time, such conditional prediction task is easy even at truly long horizon." Chaos, noise and shocks ruin it.
>
> When we say "move away from general to time-evolution paradigm": it is a precise statement. Most mainstream LTSF models investigate how $x$ transforms to $y$, so $70$-dim must compress to $30$-dim prediction; we investigate how $30$-dim noise transforms to $30$-dim prediction.
>
>
>
> 4. Formatting Choices
>
>
> "Arbitrary use of bold and italic fonts / Section 3 spans 4 pages without subsections": To comply with strict page limits while maintaining organization, we utilized bold, run-in paragraph headers (e.g., Rethinking LTSF, The Ontological Coherence of Benchmarks) to delineate distinct arguments within Section 3.
>
> Conclusion
> The review concludes with a Strong Reject (1) score, which the rubric defines as applicable when "it is impossible to tell what the nature of its contribution is." We respectfully point out that the review's opening summary acknowledges the exact nature of our contribution and notes our "strong predictive performance." We hope these clarifications resolve the misunderstandings regarding the paper's contents and definitions, and encourage the reviewer to take a fresh revisit to the substance of the paper during the discussion period.

---

> > ### Author Rebuttal · Reviewer_91rW · 2026-04-04
> >
> > I thank the authors for their response and explanations; I clarify that some of the misreadings in my reviews are due to some of the content that I consider critically relevant being only specified in the appendix and not the main body of the paper, e.g., Tables >8.
> >
> > The limitations of the paper are clear to me, but I still maintain that they are not clear in the paper. Based on the concerns raised in my review and those of the other reviewers, I don't see that they could be addressed in the paper in this rebuttal. I will increase my score by one point.

---

> > > ### Author Response · Authors · 2026-04-05
> > >
> > > We thank the reviewer for the score increase and for clarifying one source of the misreadings.
> > >
> > > When the reviewer writes: **"The limitations of the paper are clear to me, but I still maintain that they are not clear in the paper. Based on the concerns raised in my review and those of the other reviewers, I don't see that they could be addressed in the paper in this rebuttal."**
> > >
> > > We respectfully note that
> > > - We *have not heard about what the technical limitations are*, and therefore, why they *cannot* be addressed.
> > > - The reviewer's maintained confidence rating of 5 ('absolutely certain, checked math/other details carefully') appears inconsistent with the stated reason for misreading ('content that I consider critically relevant being only specified in the appendix').
> > >
> > >
> > > The 8-page limit dictated that these wide tables cannot fit in the main body and must be in the appendix. We consider it a stylistic trade-off that authors often have to make given constraint spaces. We leave this for the AC's consideration.

---

### Official Review · Reviewer_tUEd · 2026-03-12

**Soundness:** 2
**Presentation:** 2
**Significance:** 2
**Originality:** 3
**Overall Recommendation:** 3
**Confidence:** 3

**Summary:**

This paper proposes Fern, a framework for long-term time series forecasting based on ellipsoidal transport in latent space. The core idea is to model the local Jacobian of time evolution using a symmetric positive semi-definite (SPD) parameterization derived from optimal transport theory via Brenier's theorem.

**Compliance With Llm Reviewing Policy:**

Affirmed.

**Final Justification:**

Thank you for the rebuttal. My primary concern remains the practical applicability of the proposed model in real-world settings, since most of the evaluation is based on simulated data. Nevertheless, given the novel aspects of the work, I maintain a neutral overall assessment.

**Key Questions For Authors:**

see Weaknesses

**Limitations:**

see Weaknesses

**Strengths And Weaknesses:**

Strengths:
1. The paper introduces a new viewpoint that interprets forecasting as optimal transport between Gaussian distributions, with the Jacobian constrained to lie in the SPD class. This provides an interpretable geometric structure and connects time series forecasting with ideas from optimal transport and dynamical systems.
2. By parameterizing the Jacobian via spectral factors instead of arbitrary matrices, the method reduces complexity from $O(n^³)$ eigen-decomposition to $O(Rn)$ operations.

Weaknesses:
1. Although the paper invokes Brenier's theorem and optimal transport, the actual learning objective remains standard MSE regression rather than an OT-based objective. As a result, the theoretical connection between optimal transport and the learned mapping is mostly motivational rather than rigorously enforced.
2. On real datasets such as ETTh1 and ETTm1, improvements appear much smaller than on synthetic systems. This raises the question of whether the proposed method primarily benefits controlled dynamical settings rather than practical forecasting tasks.
3. The paper introduces metrics such as Wasserstein distance and EPT, which capture geometric consistency. However, most real-world forecasting tasks prioritize point prediction accuracy or probabilistic calibration. It is unclear how improvements in these geometric metrics translate into practical forecasting benefits.

---

> ### Author Rebuttal · Authors · 2026-03-26
>
> This year's ICML reviewing process is long and tiring, involving a substantial amount of work. We take this chance to sincerely thank the reviewer for the hard-work and service to the community. We address the weaknesses:
>
> "Although the paper invokes Brenier's theorem and optimal transport, the actual learning objective remains standard MSE regression rather than an OT-based objective."
>
> This is a sharp but often misunderstood point of the model.
>
> A concrete example:
> let true conditional distribution $\mathcal N(y_{true}, \Sigma)$
> A model may learn
> - $\mathcal N(y_{true}, 3*\Sigma)$ (called pred1)
> - $\mathcal N(y_{true}, \Sigma + \Sigma_{noise})$ (pred2)
>
> The learned path *is* OT from Gaussian noise to pred 1 & 2, guaranteed by the SPD structure. Since Min MSE involves matching *only* conditional expectation, the first part of $\mathcal N$, pred 1 & 2 are both **correct** for min MSE task, and they are OT *not* towards the *true* distribution, but to pred 1 & 2 resp.
>
> In short, the logic is: we use a reduced class (SPD), but can we nevertheless reach true mean and true variances with less expressivity? Brenier theorem says yes, so we know true mean is *reachable*, and **any** covariance paired with the true mean is a valid MSE-optimal solution. If our goal is to reach *true variances* on top of true mean, we'll use OT objective; for MSE-only cases, we don't need to. The instance-wise OT (from noise to any learnt prediction, such as pred 1&2) is guaranteed by SPD construction, unrelated to objective.
>
>
> "... on ETTh1 and ETTm1, improvements appear much smaller...raises the question of whether it primarily benefits controlled dynamical settings rather than practical forecasting tasks."
>
> We include ETTh1 and ETTm1 as a *completeness check* to confirm Fern remains competitive with SOTA on standard benchmarks. In our view, ETT et al. overstate their ability to reflect generalizable forecasting performance across diverse regimes:
> - **Data Artifacts**: It still suffers from zero-inflation and stuck sensors (many repeated values).
> - **Single trajectory**:  a single historical sequence with unknown dynamics and unobserved shocks, it is difficult to assess robustness beyond that trajectory.
> - **Unfalsifiable Dynamics**: since the DGP is unknown, we cannot tell if e.g. 3-5% improvements on SOTA (for *any* model) are statistically meaningful or just overfitting.
> - **Quasi-Periodicity** they are likely (since we can never confirm) a **quasi-periodic problem** as multiple frequency-only, extremely light models such as FilM and Fits perform exceptionally well and never fail catastrophically. Our view is that robustness under regime shifts, stochastic perturbations, and chaotic dynamics, which are all practical possibilities, are *not* well captured by such benchmarks, and using it to represent *all* practical forecasting risks neglecting a large class of systems with catastrophic consequences, as shown in the paper.
>
> The deeper question is, do ETTh1 and ETTm1 *represent* practical forecasting tasks to the degree that SOTA on ETT translates to better general performances? Ma et al. (2026) (https://arxiv.org/abs/2602.01736) surveyed the field and concluded that in domains such as weather, energy and electricity (in the **same domain** as the benchmarks), academics and practitioners develop their own methods with built-in prior knowledge rather than general models that are SOTA on these benchmarks. The universal rejection in the past 2-3 years suggests we should reconsider their value.
>
> "... metrics such as Wasserstein distance and EPT, which capture geometric consistency. However, most tasks prioritize point prediction accuracy or probabilistic calibration. It is unclear how these translate into benefits"
>
> We saw how geometry can be surprisingly predictive. In Fig.4, eigenvalue diagnostics predicted where Fern would fail on Lorenz-63 (overconfident low-variance regions before lobe switches).
>
> WD: for example, while stock returns of [5,3,-8] and [0,1,-1] both have the same mean, the downward momentum *is* important for high frequency trading. Also, the positive and negative signs have different meanings: one can lose at most 100%, while the upside is unbounded; for the same conditional mean, the comprehensive downside risk *is* a geometric property: how long, how far, and how frequently do earnings dip below zeroes.
>
> MSE paired with geometric metrics aligns *more* with human intuition of "pointwise accuracy" -- not just expectation, but the shape of two series has to match. MSE alone provides too little information.
>
> EPT: this is used to establish the natural upper end of the prediction models. Currently, the choice of 96, 192, 336 and 720 is arbitrary and rightfully criticized in Bergmeir's article cited in the paper. EPT helps the reader to establish a bound: if EPT is about 230 for all models in a 336 prediction, then the first 230 should be treated more seriously, and the latter as exploratory.

---

> > ### Author Rebuttal · Reviewer_tUEd · 2026-04-04
> >
> > Thank you for the clarification, but I am still not fully convinced.
> >
> > I agree that standard benchmarks such as ETTh1 and ETTm1 have limitations and may not fully reflect robustness under regime shifts or stochastic dynamics. However, these limitations alone do not justify dismissing weaker improvements on them. Since they are still widely used as standard forecasting benchmarks, the relatively smaller gains on these datasets still raise the question of whether Fern’s advantages are more pronounced in controlled dynamical settings than in conventional practical forecasting tasks.
> >
> > In addition, the statement that “the universal rejection in the past 2–3 years suggests we should reconsider their value” remains too broad and insufficiently supported. The authors should clarify what is meant by “universal rejection” and provide more concrete evidence for this claim.

---

> > > ### Author Response · Authors · 2026-04-04
> > >
> > > We thank the reviewer for the follow-up. We realize our previous response did not adequately address your core question due to space constraints, and we sincerely appreciate the opportunity to engage further on this critical topic.
> > >
> > > **The reviewer wrote: "I agree that standard benchmarks such as ETTh1 and ETTm1 have limitations ... However, these limitations alone do not justify dismissing weaker improvements on them. Since they are still widely used ... the relatively smaller gains on these datasets still raise the question of whether Fern’s advantages are more pronounced in controlled dynamical settings than in conventional practical forecasting tasks."**
> > >
> > > Table 2 shows that, Fern's weaker advantages and occasional disadvantages, are not limited to ETTh1 and ETTm1: For example, on SLDS-Base, mtcn's MSE is 1.96 vs Fern's 2.84 (where TimeMixer is 4.54 and DLinear is 4.42). Fern's disadvantage is larger than in the case of ETTh1 (10.97 vs 10.39). On SAR-Base, Fern is on par with TimeMixer and Modern TCN with 0.55, and the three share the first place.
> > >
> > > This leads to our **first observation**: modern TSF methods are doing quite well on *base/generic* stochastic systems. Fern's *main advantage* is on *non-stationary shocks*: from SLDS-Base to SLDS-Switch, Fern's MSE is 2.84->4.05 while patchtst is 2.27(a better score!)->9.56. The MSE differences across models on ETTh1 and m1 appears to be consistent with the base stochastic systems; in fact, the gaps are smaller.
> > >
> > >
> > > Ever since Autoformer and ETSFormer, many researcher **explicitly** introduce trend+seasonal cycle+noise priors, *modeling* the ETT's as a SAR (seasonal autoregressive) or SARIMA processes implicitly. We pose this question: **if ETT is *treated like* a SAR process, why don't we go one step further, directly test the SAR system**? Indeed, error bands across models on ETT's are as tight as in the SAR-base case. When we do that, different seeds generate *different trajectories* under different *explicit* shocks. Here is a test over 4 seeds under the same shock specs in the paper:
> > >
> > > | Dataset      | fr-64 | fr-192 | best non-fr-64 | best non-fr-192 |
> > > |--------------|------:|-------:|---------------:|----------------:|
> > > | SLDS-Base    | 1.00  | 1.06   | 1.03 (tfm)     | 1.15 (tfm)      |
> > > | SLDS-Switch  | 1.20  | 1.31   | 1.24 (tfm)     | 1.43 (tfm)      |
> > > | SAR-Base     | 0.144 | 0.134  | 0.146 (tfm)    | 0.150 (c2)      |
> > > | SAR-Param    | 0.343 | 0.336  | 0.363 (tfm)    | 0.377 (tfm)     |
> > > | DW-Base      | 0.135 | 0.144  | 0.132 (tfm)    | 0.132 (c2/tfm)  |
> > > | DW-Switch    | 0.322 | 0.504  | 0.320 (cb)     | 0.513 (cb)      |
> > >
> > > The shown probabilistic metric is CRPS, and the other models tested includes Chronos2, TimeFM and Chronos-Bolt, all foundational large generative models capable of generating probabilistic forecasts. Across base stochastic systems (SLDS-Base, SAR-Base, DW-Base), Fern is competitive but not uniformly dominant. Under switched variants, however, the relative picture changes: on DW-Switch, Fern moves from near-parity to substantial advantages, especially at the longer horizon. Over more seeds, we can even conclude what are the best/average/worst scenarios when the real-world dataset is *this system with these shocks*, and even if the system and the model are **statistically** close enough at e.g. p-value = 0.05.
> > >
> > > If SAR sounds too simple or naive, modern synthetic benchmarks such as SynTSBench (NeurIPS 2025, https://arxiv.org/abs/2510.20273) and ARIES (https://arxiv.org/abs/2509.06060) are ready to help **generating a system**, not a single trajectory, that is calibrated to linealy or temporally look like ETT. We conclude with our **second observation**: ETTh1/m1 look like and have been treated like an SAR system with slight, slow trends. If so, using a genuine **system** makes a much stronger case for benchmarking.
> > >
> > > - **the statement that “the universal rejection in the past 2–3 years suggests we should reconsider their value” remains too broad and insufficiently supported.**
> > >
> > >
> > > We agree 'universal rejection' was too strong; we were paraphrasing the position paper Ma et al. (NeurIPS 2026) (https://arxiv.org/abs/2602.01736) which states "As a result, network architectures developed aiming at fitting general time series domains are almost not inspiring for real world practices for certain single (or few similar) domains such as Finance, Weather, Traffic, etc: **each specific domain develops their own methods that rarely utilize advances in neural network architectures of time series community in recent 2-3 years.** See Table 1 and 2 for a summary, and we invite the reviewer to browse section 5 for an outline of the actual patterns being adopted in those specific domains -- PINN for weather, and e.g. meta-learning models etc. Our statement is about the **implied generalizability claim** "better ETT/Weather performances -> better Energy/Climate performance", not about the datasets themselves.

---

### Official Review · Reviewer_JSTS · 2026-03-13

**Soundness:** 3
**Presentation:** 3
**Significance:** 3
**Originality:** 3
**Overall Recommendation:** 5
**Confidence:** 3

**Summary:**

This paper proposes Fern, a novel framework that conceptualizes long-term time series forecasting as the optimal transport of probability mass from a fixed Gaussian source to data-dependent ellipsoids. Grounded in Brenier’s theorem, Fern directly parameterizes the local Jacobian as a Symmetric Positive Semi-Definite (SPD) factorization, which allows the model to learn the spectral structure—eigenvalues and eigenvectors—of the underlying manifold with linear rather than cubic computational complexity. This geometric approach provides interpretable diagnostics and superior robustness against non-stationary shocks, stochastic noise, and deterministic chaos. To address the limitations of existing benchmarks, the authors introduce a synthetic stress-testing environment and the Effective Prediction Time (EPT) metric. Experimental results show that Fern significantly outperforms state-of-the-art baselines like DLinear and Koopa, exhibiting up to two orders of magnitude greater stability in regime-shifting scenarios while maintaining high computational efficiency through parallel patch-wise processing.

**Compliance With Llm Reviewing Policy:**

Affirmed.

**Key Questions For Authors:**

The method is built on the fact that the Wasserstein-2 optimal map between Gaussian distributions is affine, which motivates the SPD Jacobian parameterization and the ellipsoidal transport view. However, many real forecasting targets may be skewed, heavy-tailed, or multi-modal, particularly under regime switches or event-driven shocks. Could the authors clarify whether Fern should be viewed primarily as a strong Gaussian approximation method, or whether they believe the geometric transport mechanism remains effective even when the target conditional distribution is substantially non-Gaussian? Some discussion or evidence on the sensitivity of Fern to violations of this assumption would help assess the scope of the method.

Could the authors elaborate on when patching is expected to be harmless versus harmful, and whether there are failure modes where the shared backbone is insufficient to recover dependencies across distant forecast segments?

The reported gains in Table 2 are striking, especially on chaotic and shock-driven systems, but the paper also argues that benchmark outcomes can be highly sensitive to training protocol details such as early stopping and grace periods. In addition, the model capacity depends on design choices such as the number of Householder reflections and patch granularity. Could the authors provide more evidence that Fern’s advantage is stable under fair and comparable tuning budgets across baselines, rather than benefiting disproportionately from method-specific settings?

**Limitations:**

The patch-wise parallel transport design is computationally attractive, but it may underrepresent long-range interactions across forecast segments. The paper explicitly frames forecasting as patchwise geometric transport, and the ablations suggest that patching is an important part of the design. However, this decomposition raises the concern that dependencies spanning multiple patches may be delegated to the shared encoder/backbone rather than modeled directly in the transport step itself. This could limit performance in settings where cross-patch coupling is central to the dynamics.

The method appears somewhat sensitive to architectural and training choices. The ablations vary the number of Householder reflections and the presence of patching/rotation, indicating that these design choices do affect performance. In addition, the experimental setup uses different validation and training configurations across the “shock table” and “detailed table,” including different learning rates, batch sizes, grace periods, and even WD versus projected SWD evaluation variants. Although these choices are disclosed, they make it harder to assess how robust the reported gains are to tuning decisions and how easily the method can be deployed without careful calibration on new datasets.

**Strengths And Weaknesses:**

Strengths
The paper is grounded in a clear structural motivation and offers interpretable diagnostic signals beyond point forecasts. By tracking quantities such as the spectral radius and trace as indicators of geometric expansion/contraction, the framework provides a meaningful lens into where and why the model expects forecasting to be difficult, not just what it predicts. The work also reflects thoughtfully on current LTSF evaluation practice and augments it with distribution- and reliability-oriented metrics, introducing WD/SWD and, in particular, EPT as an intuitive threshold-based measure of when predictions become unreliable—especially relevant for chaotic dynamics. Finally, the stress-test results are striking: Table 2 reports that under several chaotic or shock-driven settings, Fern can outperform strong baselines such as DLinear and Koopa by up to two orders of magnitude in error. If these gains hold under rigorous reproducibility and fair hyperparameter tuning, they provide compelling evidence of the method’s robustness under challenging non-stationary regimes.

Weaknesses:
1. Assumption of Gaussianity: The core formulation relies on the fact that the unique W2 optimal transport map between two Gaussian distributions is strictly affine. While this simplifies the math and ensures an SPD Jacobian, real-world data may follow highly non-Gaussian or multi-modal distributions where a single ellipsoidal (Gaussian) approximation per patch might be too restrictive.
2. Patch-wise Independence: The method utilizes parallel patch-wise predictive transport. While this is computationally efficient and helps avoid the "curse of dimensionality," it may potentially overlook complex inter-patch dependencies or long-range temporal correlations that cannot be fully captured by the shared backbone alone.
3. Hyperparameter Sensitivity: The performance and capacity of the model are tied to hyperparameters such as the number of Householder reflections (R) and the patch size (p). Determining the optimal balance between computational speed (low R) and model expressivity (high R) may require significant tuning for different types of datasets.

---

> ### Author Rebuttal · Authors · 2026-03-27
>
> We sincerely thank the reviewer for the comprehensive review and insightful questions.
>
> **"Assumption of Gaussianity" and Q1**: Empirically, we considered two general cases: (1) we tested an MOE version of Fern, with both soft and hard routing, which results in a Gaussian mixture model that can handle multi-modal distributions; (2) we tested a nonlinear version of Fern, replacing the SPD matrix in the OT head with the Cascaded Network (https://arxiv.org/pdf/2301.10862), where we can stack additional monotone activations while still yielding a SPD Jacobian.
>
> We found:
> - the MOE and C-MGN version provides little improvements on the tested datasets
> - The eigenvalues can still be directly calculated, but the spectral structures are not as straightforward
> - Due to space constraint and the philosophy to make models simpler, we didn't include the material, we can add discussions in the final version.
>
> In short: Fern is very flexible and has two natural extensions to handle situations where target is substantially non-Gaussian. Since Brenier's theorem doesn't require target measure $\nu$ to be even abs. cont., our choice of Gaussian assumption is to simplify the presentation and math, as the reviewer correctly pointed out, rather than an inherent limitation to the approach.
>
> **"Patch-wise Independence" and Q2**: We think the cross patch dependency can be quantified: in Table 3 and 8, "no patching" means $y_1$ is allowed to communicate with even $y_192$ in a horizon of length 192, which means arbitrary cross patch dependency is allowed. Its worse performances against baseline suggest that long-range dependency *in this setting* doesn't pose a serious problem.
>
> Another reason why we are less concerned about cross patch coupling is that (1) the shared encoder conditions each patch on the full input, so cross-patch dependencies are already encoded before the transport step; (2) whatever the coupling is, they must materialize into *some* distribution $y$; when we set # of Householder reflections R to be equal P, the patch size, this ensures that $y$ is *searchable* by our SPD machinery. Indeed, in table 8, "no patching" variant is superior to $patching with R=2$ variant, suggesting $R$ is a more critical bottleneck.
>
> The reviewer also provides a thoughtful hypothesis: indeed, while the shared MLP backbone distills information of the entire $x$ to each patches, *it is possible* that Transformers (with $O(n^2)$ pairwise understanding of time points) and RNN with autoregressive rollouts may provide a better *hidden* representation. This means treating OT heads as a *wrapper* to Transformer or RNN: instead of directly output $y$, in $y$-space we only allow smooth, well-behaved and structure-rich SPD transformations *based on* RNN and others. This is really an interesting area for further exploration and we thank the reviewer for this valuable insight.
>
> **"Hyperparameter Sensitivity" and Q3**: The shock table has grace period=3 and detailed table has grace period=0, providing a comparison under different grace periods. All baselines share identical optimizer, LR, batch size, and early-stopping protocol within each table. As for the concern about tuning needs for Householder reflections, our experience is that performance is stable across a wide range of $R$ and $R=8$ is a sweet spot for the tested datasets. To ease the worry further, we designed two methods to make it easy to simply use the *maximum number of R*:
>
> (1) block update: for example, we can use 24 reflections but only update 8 of them during a single pass, saving memory. On Lorenz63, 25k samples, we don't see a large difference in training time and performances:
>
> Reflection: 8
> MSE=1.109 | MAE=0.570
> time: 484.86 seconds
>
> &nbsp;
>
> Reflection: 24 train with 3 size-8 blocks (updates one third each forward)
> MSE=1.286 | MAE=0.628
> time: 457.08 seconds
>
> (2) (implemented in code) we write Triton code to use this trick: since every Householder reflection is involutory ($H^2 = I$, i.e. $H = H^{-1}$), applying the same reflection to $x_r$ recovers $x_{r-1}$. We can save the final output and apply reflection in reverse: instead of saving R intermediate vectors per patch ($O(R*P)$ memory), we do $R$ extra reflection applications during backward (just a dot product and a fused multiply-add per step). Memory cost stays $O(P)$ per patch regardless of R.
>
> Since Householder reflection is cheap and the bottleneck is in the memory usage during backprop, these two methods allow us to reduce memory usage and increase the number of reflections, at the expense of slower update *or* extra reflections. We can simply set R=P (patch size) without tuning. The only failure mode is when P is set too small that the intrinsic dim is larger than P; in the paper, systems like Lorenz has 3 dims, and Takens theorem works very well with delay-embedding with >10 dims. We use a conservative 24 dim as default, empirically setting 48 dim yields little performance differences.

---

> > ### Author Rebuttal · Reviewer_JSTS · 2026-04-03
> >
> > Thank you for your response, I will keep my score.

---

### Decision · Program_Chairs · 2026-04-30

**Decision:**

Accept (regular)

**Comment:**

Reviewers broadly recognized Fern's originality. The paper constrains forecasting dynamics through an SPD Jacobian parameterization motivated by Brenier's theorem, giving both a geometric interpretation and spectral diagnostics such as radius and trace. The Householder-based parameterization gives a credible efficiency story. Reviewers also valued the stress-test suite and the Effective Prediction Time metric, which probe regime shifts and chaotic dynamics more directly than standard long-horizon benchmark tables.

The main reservations were also fairly consistent. First, the OT connection is viewed by some reviewers as more motivational than operational, since the model is ultimately trained with Huber/MSE rather than an OT objective. Second, the strongest gains appear on controlled synthetic systems, while improvements on standard real datasets such as ETTh1 and ETTm1 are smaller and sometimes absent, which weakens any broad claim of practical superiority. Third, one reviewer in particular felt the paper’s presentation fell below conference standard: too informal, poorly organized, and at points hard to follow, though some of the specific complaints were later softened after the rebuttal clarified that several missing details were in fact present, often in the appendix. The post-review discussions also confirmed the shared concerns across the reviewers.

Overall, I think the idea is interesting and novel enough, but it's a pity that the paper didn't go all the way with OT as the objective, without which, I see it as merely a similar construct as a multivariate regression model x \mapsto \mu(x), \Sigma(x) with \Sigma(x) being parameterized with eig (HH parameterization of Q is standard). On top of it, the presentation needs to be improved, with boldface letters everywhere doing a disservice to "highlight" the important ideas. I'd vote for weak acceptance, slightly above the bar, but not much higher.